# Atypical memory B-cells are associated with *Plasmodium falciparum* anemia through anti-phosphatidylserine antibodies

Juan Rivera-Correa[1], Maria Sophia Mackroth[2,3,4], Thomas Jacobs[3,4], Julian Schulze zur Wiesch[2,4], Thierry Rolling[2,4,5], Ana Rodriguez[1]*

[1]Department of Microbiology, New York University School of Medicine, New York, United States; [2]Division of Infectious Diseases, I. Department of Internal Medicine, University Medical Center Hamburg-Eppendorf, Hamburg, Germany; [3]Protozoa Immunology, Bernhard Nocht Institute for Tropical Medicine, Hamburg, Germany; [4]German Center for Infection Research (DZIF), Hamburg-Lübeck-Borstel-Riems, Hamburg, Germany; [5]Department of Clinical Research, Bernhard Nocht Institute for Tropical Medicine, Hamburg, Germany

**Abstract** Anemia is a common complication of malaria that is characterized by the loss of infected and uninfected erythrocytes. In mouse malaria models, clearance of uninfected erythrocytes is promoted by autoimmune anti-phosphatidylserine (PS) antibodies produced by T-bet[+]B-cells, which bind to exposed PS in erythrocytes, but the mechanism in patients is still unclear. In *Plasmodium falciparum* patients with anemia, we show that atypical memory FcRL5[+]T-bet[+] B-cells are expanded and associate both with higher levels of anti-PS antibodies in plasma and with the development of anemia in these patients. No association of anti-PS antibodies or anemia with other B-cell subsets and no association of other antibody specificities with FcRL5[+]T-bet[+] B-cells is observed, revealing high specificity in this response. We also identify FcRL5[+]T-bet[+] B-cells as producers of anti-PS antibodies in ex vivo cultures of naïve human peripheral blood mononuclear cells (PBMC) stimulated with *P.-falciparum*-infected erythrocyte lysates. These data define a crucial role for atypical memory B-cells and anti-PS autoantibodies in human malarial anemia.
DOI: https://doi.org/10.7554/eLife.48309.001

*For correspondence:
ana.rodriguez@nyumc.org

Competing interests: The authors declare that no competing interests exist.

## Introduction

Malaria is still a major global health threat with over 200 million new infections and around 400,000 deaths in 2017 (*World Health Organization, 2018*). Anemia is a common complication associated with malaria that contributes significantly to the great morbidity and mortality associated with the disease (*White, 2018*). Despite its high clinical relevance, the mechanisms underlying malarial anemia in patients remain largely unknown. The difficulty in studying this syndrome arises at least in part from its multi-factorial etiology, as malaria causes both the clearance (through complement-mediated lysis or phagocytosis) of infected and uninfected erythrocytes and bone marrow dyserythropoiesis (*Lamikanra et al., 2007*; *White, 2018*). The clearance of uninfected erythrocytes is thought to contribute significantly to anemia, because for each erythrocyte lysed directly due to *Plasmodium* infection, about eight uninfected erythrocytes are killed in *P. falciparum* infections (*Jakeman et al., 1999*; *Price et al., 2001*) and about 34 erythrocytes are killed in *P. vivax* infections (*Collins et al., 2003*).

The anti-parasite B-cell antibody response that is generated during the malaria blood stage represents an essential component of the protective immune response against this disease (*Doolan et al., 2009*; *Portugal et al., 2013*). However, an important autoimmune response is also generated during malaria, in which autoantibodies mediate some of the associated pathologies (*Hart et al., 2016*; *Rivera-Correa and Rodriguez, 2018*; *Rivera-Correa, 2016*). Specifically, autoantibodies that target membrane phosphatidylserine (PS) on uninfected erythrocytes promote anemia during malaria in mouse models, and these autoantibodies correlate with low hemoglobin levels in a cohort of *P.-falciparum-* and *P.-vivax-*infected patients with malarial anemia (*Barber et al., 2019*; *Fernandez-Arias et al., 2016*), establishing an autoimmune component to malarial anemia. An atypical group of B-cells that express the transcription factor T-bet secrete autoantibodies in different autoimmune mouse models (*Rubtsova et al., 2015*; *Rubtsova et al., 2017*) and were found to be major producers of anti-PS antibodies in mouse malaria (*Rivera-Correa et al., 2017*). The relation of these atypical T-bet[+] B-cells to autoantibodies and their role in malarial anemia has not been assessed in malaria patients.

B-cells expressing the transcription factor T-bet have been identified in the circulation of individuals from malaria-endemic areas and implicated in the memory response against *Plasmodium* (*Guthmiller et al., 2017*), being considered atypical memory B-cells (*Obeng-Adjei et al., 2017*; *Weiss et al., 2009*). These cells are characterized as secreting low levels of antibodies and by inhibitory phenotypic markers such as FcRL5 (*Sullivan et al., 2015*). The secreted-antibody specificity of these atypical memory B-cells has not been characterized because of their reduced effector function and minimal antibody secretion in-vitro (*Portugal et al., 2015*). Hence studying these FcRL5[+]T-bet[+] B-cells during acute stage malaria in a population from a non-endemic area, such as European travelers, represents a unique opportunity to study the antibody specificity that results from recent primary activation.

In this study, we focused on measuring the levels of atypical FcRL5[+]T-bet[+] B-cells in the circulation of *P.-falciparum-*infected returned travelers and its relation to autoantibodies and anemia development. Our results show that atypical FcRL5[+]T-bet[+] B-cells are greatly expanded in acute malaria in *P.-falciparum-*infected patients and correlate with both anemia development and plasma anti-PS antibody levels in these patients. In vitro studies confirmed that the activation of FcRL5[+]T-bet[+] B-cells by *P.-falciparum-*infected erythrocytes induces the secretion of anti-PS autoantibodies. All together, these findings attribute a role to atypical FcRL5[+]T-bet[+] B-cells and anti-PS antibodies in the pathogenesis of human malarial anemia. Targeting these cells could have a therapeutic benefit in treating anemia during malaria.

## Results

### Specific autoantibodies correlate with malarial anemia and erythrocyte lysis capacity in *P.-falciparum*-infected patients

In this study, we focused on samples (24 patients, 31 unique samples) from a cohort of *P.-falciparum-*infected returned travelers from Germany, who acquired malaria while visiting Africa. This cohort suffered from mild anemia with average hemoglobin levels of 12.4 g/dL (males) and 10.2 g/dL (females) (normal range is 13.8 to 17.2 and 12.1 to 15.1 g/dL, respectively) (*Table 1*).

As described before in other cohorts with mild anemia (*Fernandes et al., 2008*; *Sumbele et al., 2016*), hemoglobin levels in this cohort do not significantly correlate with parasitemia (*Figure 1A*), confirming that direct erythrocyte infection by *Plasmodium* is not a major cause of anemia and indicating that other mechanisms must contribute to this pathology. This is in agreement with previous findings reporting major losses of uninfected erythrocytes and dyserythropoiesis during malaria (*White, 2018*).

As we had previously observed that autoimmune anti-PS antibodies induce anemia during malaria in a mouse model (*Fernandez-Arias et al., 2016*; *Rivera-Correa et al., 2017*), we determined whether hemoglobin levels correlated with autoimmune anti-PS IgG antibodies in our cohort. We observed an inverse correlation between anti-PS antibodies and hemoglobin levels (*Figure 1B*), which was not found for IgG antibodies against the *P. falciparum* erythrocyte binding antigen (PfEBA) (*Figure 1C*), suggesting that an autoimmune response contributes to the development of anemia in malaria.

**Table 1.** Clinical information from *P.-falciparum*-infected returned German travelers.

| Subject ID | Day of sampling[&] | Hemoglobin (g/dl)* | Hemoglobin (g/dl)** | Thrombocyte count (1000/μl)* | Parasite count/μl **, [#] | Red blood cell (RBC) count (million/μl)** | Sex | Age | Type of patient[$] | Country of infection |
|---|---|---|---|---|---|---|---|---|---|---|
| 100 | 3 | 10.3 | 13.4 | 135 | 132,000 | 4.6 | m | 51 | VFR | Nigeria |
| | 31 | ND | 13.4 | ND | 132,000 | 4.6 | m | 51 | VFR | Nigeria |
| 101 | 8 | 7.3 | 12.1 | 270 | 1,860,000 | 3.89 | f | 56 | T | Gambia |
| | 24 | 8.3 | 12.1 | 589 | 1,860,000 | 3.89 | f | 56 | T | Gambia |
| 102 | 6 | 13.5 | 16.2 | 140 | <52,800 | 5.28 | m | 63 | T | Uganda |
| | 27 | ND | 16.2 | ND | <52,800 | 5.28 | m | 63 | T | Uganda |
| 103 | 2 | 13.8 | 14.4 | 54 | 46,900 | 4.69 | m | 38 | VFR | Guinea |
| 104 | 3 | 12.6 | 13.6 | 130 | 470,000 | 4.7 | m | 56 | T | Madagascar |
| | 7 | 13.7 | 13.6 | 321 | 470,000 | 4.7 | m | 56 | T | Madagascar |
| 105 | 5 | 7.8 | 8.1 | 135 | 1,050,000 | 3.5 | f | 53 | VFR | Kenya |
| 106 | 2 | 10.8 | 11.9 | 9 | 63,900 | 4.26 | m | 43 | VFR | Ghana |
| 107 | 3 | 11.6 | 12.4 | 48 | >430,000 | 4.3 | m | 52 | VFR | Ghana |
| | 12 | 11.4 | 12.4 | 475 | >430,000 | 4.3 | m | 52 | VFR | Ghana |
| 108 | 0 | 10.3 | 10.3 | 156 | 176 | 3.83 | m | 50 | VFR | Benin |
| 109 | 2 | 10.5 | 10.8 | 98 | 16 | 3.73 | m | 62 | VFR | Ghana |
| 110 | 3 | 12.3 | 13.5 | 128 | 366,800 | 5.24 | f | 20 | VFR | Tanzania |
| 111 | 1 | 13.5 | 14.4 | 23 | 144,600 | 4.82 | m | 35 | T | Nigeria |
| 112 | 3 | 13.1 | 13.7 | 70 | 160,200 | 5.34 | m | 26 | VFR | Benin |
| 113 | 4 | 10.7 | 13.7 | 74 | 340,000 | 4.25 | m | 39 | VFR | Unknown |
| 114 | 3 | 16.6 | 18.7 | 53 | 4840 | 5.96 | m | 26 | T | Ghana |
| 115 | 0 | 12.6 | 12.6 | 53 | 14,762 | 4.84 | m | 62 | VFR | Ghana |
| 116 | 2 | 12.5 | 12.9 | 80 | 26,917 | 4.58 | f | 43 | VFR | Cameroon |
| | 4 | 10.4 | 12.9 | 106 | 26,917 | 4.58 | f | 43 | VFR | Cameroon |
| 117 | 4 | 15 | 19.5 | 32 | 492,800 | 6.16 | m | 46 | T | Nigeria |
| 118 | 1 | 11 | 12.1 | 20 | 296,100 | 4.23 | m | 39 | T | Uganda |
| 119 | 1 | 11.9 | 11.9 | 123 | 29,800 | 4.02 | f | 25 | VFR | Ivory Coast |
| | 3 | 11.6 | 11.9 | 116 | 29,800 | 4.02 | f | 25 | VFR | Ivory Coast |
| 120 | 0 | 16.1 | 16.1 | 102 | 7896 | 5.36 | m | 38 | VFR | Guinea Bissau |
| 121 | 2 | 14.4 | 15.7 | 119 | 496 | 5.48 | m | 34 | VFR | Nigeria |
| 122 | 2 | ND | 13.5 | ND | 74 | 5.15 | m | 18 | VFR | Togo |
| 123 | 2 | 12.8 | 14.2 | 60 | 49,800 | 4.98 | m | 52 | VFR | Ghana |

[&]Days since treatment start to sampling, *Measurement at day of sampling, **Measurement at day of presentation, [#]Parasitemia expressed in infected erythrocytes per μl of blood, [$]Tourist (T), Visiting Friend or Relative (VFR). ND, not determined.

DOI: https://doi.org/10.7554/eLife.48309.002

To further dissect the role that autoantibodies could be playing during malarial anemia in *P.-falciparum*-infected patients, we analyzed other relevant autoantibodies: anti-erythrocyte and anti-DNA IgG antibodies. As expected, our results show that plasma anti-erythrocyte IgG antibodies also correlate with the development of malarial anemia, showing a significant negative correlation with hemoglobin from the samples of *P.-falciparum*-infected returned travelers (*Figure 1D*). These antibodies recognized all kinds of antigens in human erythrocyte lysates (*Mourão et al., 2018*; *Mourão et al., 2016*), including PS. The similarities in the correlations between anti-PS and anti-erythrocyte IgG antibodies and anemia may indicate that anti-PS antibodies are the major antibody specificity driving anemia.

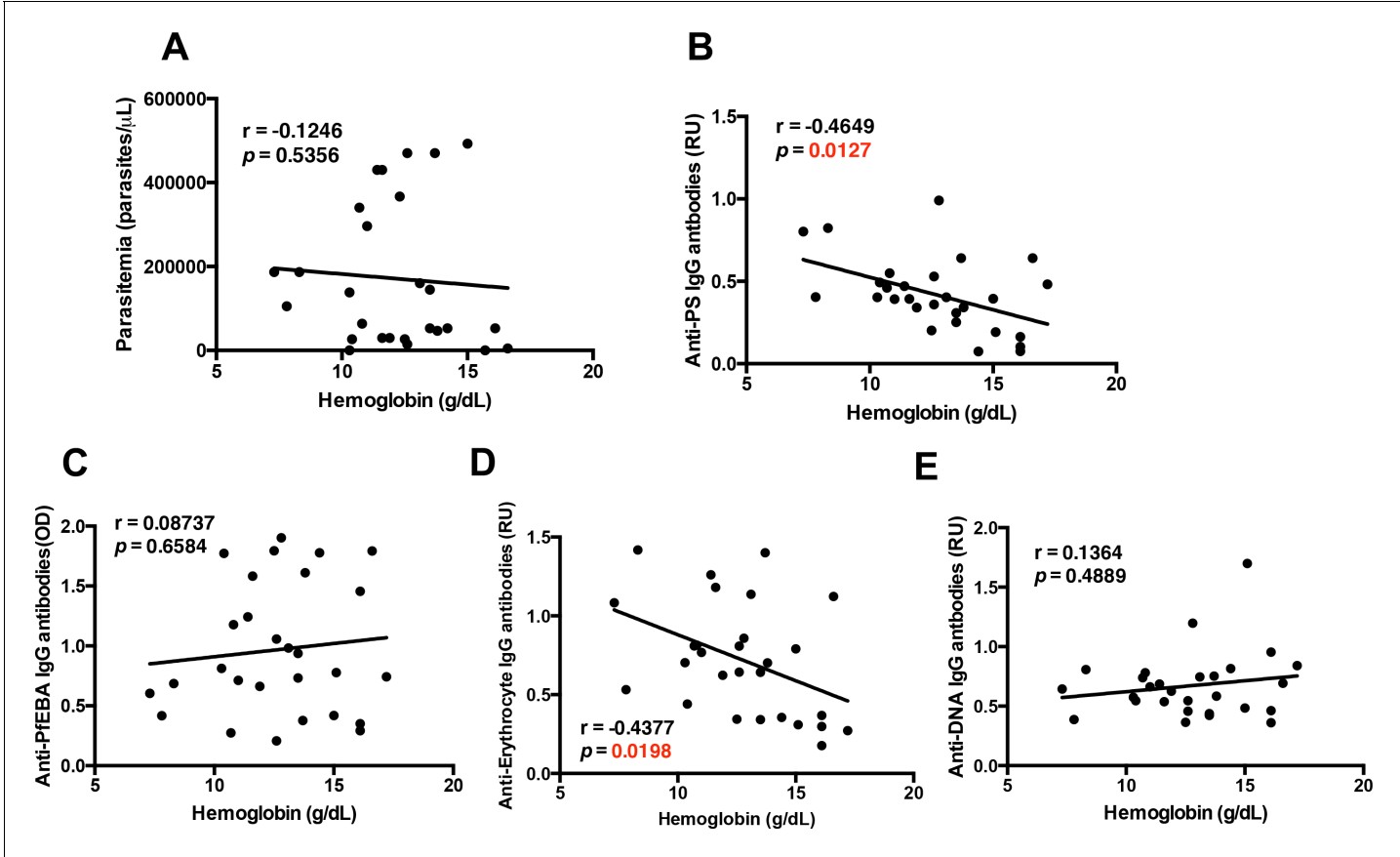

**Figure 1.** Specific autoantibodies correlate with malarial anemia in *P.-falciparum*-infected returned travelers. Non-parametric Spearman correlation analysis comparing hemoglobin with (**A**) parasitemia, (**B**) anti-PS IgG antibodies, (**C**) anti-PfEBA IgG antibodies, (**D**) anti-erythrocyte IgG antibodies and (**E**) anti-DNA IgG antibodies.

DOI: https://doi.org/10.7554/eLife.48309.003

The following source data is available for figure 1:

**Source data 1.** Source data for *Figure 1* .
DOI: https://doi.org/10.7554/eLife.48309.004

In addition, our results show that anti-DNA IgG antibodies, which are typical in autoimmune diseases (*Tsokos, 2011*), do not correlate with anemia (*Figure 1E*), suggesting that the specificity of the autoimmune antibody response is important for the development of anemia. Collectively, these initial correlations point to an autoimmune origin of malarial anemia, and confirm that the cohort of *P.-falciparum*-infected returned travelers is an adequate study group for testing the contribution of atypical B-cells in malarial anemia.

As malarial anemia is characterized by the lysis of uninfected erythrocytes and by an increase in lactate dehydrogenase (LDH) in the plasma (*Fendel et al., 2010*), we determined whether anti-PS IgG antibodies correlate with the levels of LDH in the patient's plasma (*Figure 2A*). We observed a significant positive correlation, which suggests that autoimmune anti-PS IgG antibodies may induce the lysis of erythrocytes during malaria. We next determined whether erythrocyte lysis capacity was increased in the plasma of *P.-falciparum*-infected patients. Using an in vitro complement lysis assay of human erythrocytes exposing PS, we observed that plasma from patients was significantly more effective than healthy control plasma in lysing human erythrocytes (*Figure 2B*).

We then studied the relation of anti-PS IgG levels and the erythrocyte lysis capacity in the patient's plasma, as determined using the in vitro complement lysis assay. We observed a direct correlation between anti-PS and erythrocyte lysis capacity (*Figure 2C*), which suggests that anti-PS IgG

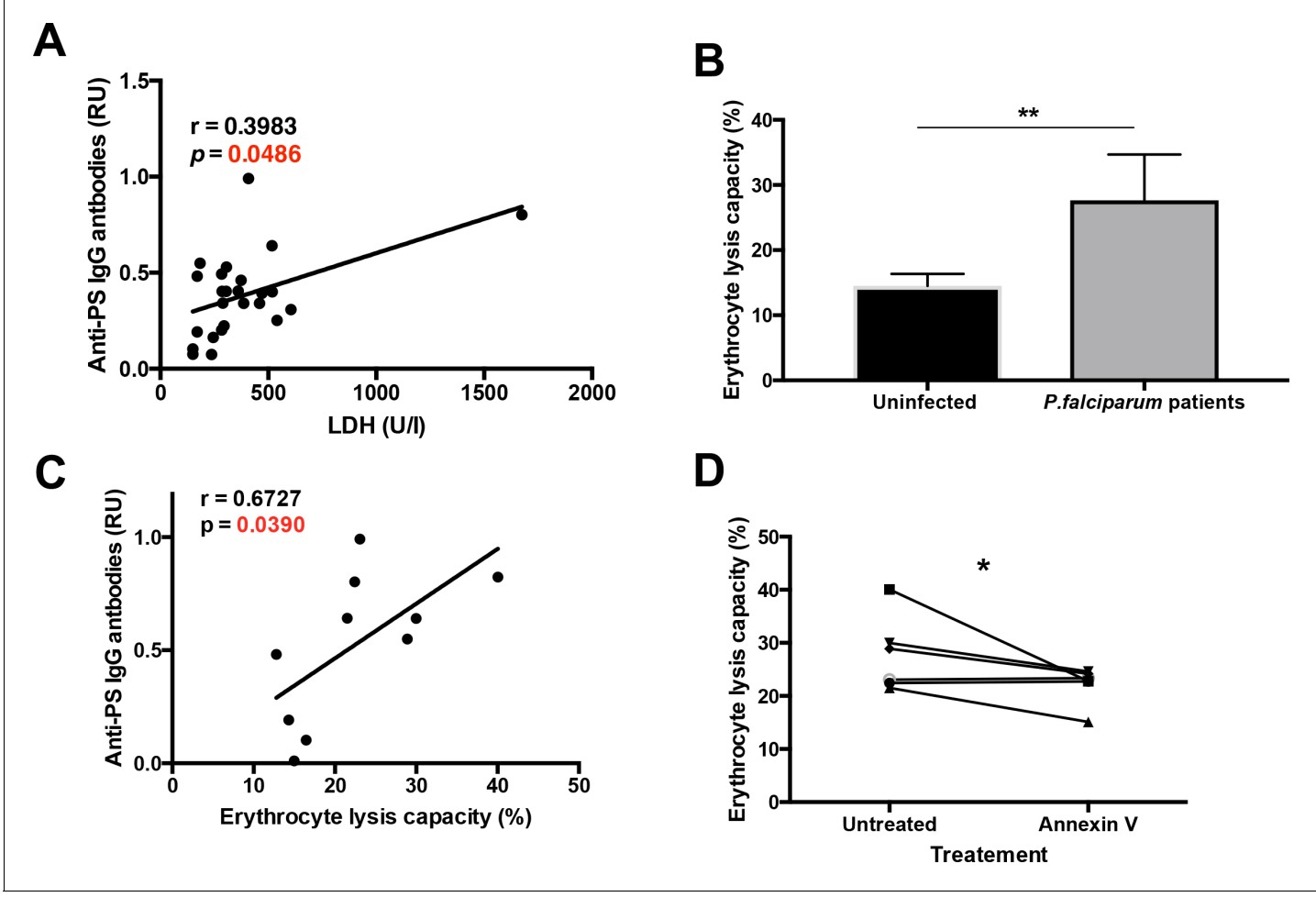

**Figure 2.** Plasma from *P. falciparum* patients mediates erythrocyte lysis, which can be partially inhibited by Annexin V. (**A,B**) Correlation of plasma anti-PS IgG antibodies with the LDH levels (**A**) or with the erythrocyte lysis capacity (**B**) of the plasma of *P. falciparum* patients. (**C**) Complement-mediated lysis of erythrocytes exposing PS by *P. falciparum* patient's plasma compared to plasma from uninfected controls, expressed as percentage of maximal lysis. (**D**) Complement-mediated lysis of erythrocytes exposing PS, pre-incubated or not with Annexin V, before incubation with the plasma of *P. falciparum* patients (n = 6). Results show the means and standard deviations of triplicated determinations. Significance was assessed by nonparametric Spearman correlation analysis (**A,B**) or unpaired Student's t-test (**C,D**). *p≤0.05, **p≤0.01.
DOI: https://doi.org/10.7554/eLife.48309.005

The following source data is available for figure 2:

**Source data 1.** Source data for *Figure 2*.
DOI: https://doi.org/10.7554/eLife.48309.006

antibodies may contribute to anemia in malaria by inducing the complement-mediated lysis of uninfected erythrocytes.

To determine the anti-PS specificity of the erythrocyte lysis, we pre-incubated the erythrocytes with annexin V, a protein that specifically binds to PS and inhibits the binding of anti-PS antibodies (*Fernandez-Arias et al., 2016*; *van Engeland et al., 1998*), finding a partial reduction of the erythrocyte lysis capacity in the plasma samples (*Figure 2D*). It is likely that other antibody specificities (*Mourão et al., 2018*; *Mourão et al., 2016*) in addition to anti-PS also contribute to erythrocyte lysis in malaria patients. Taken together, these results suggest that anti-PS antibodies mediate the lysis of uninfected erythrocytes that expose PS during malaria.

## Atypical memory FcRL5$^+$T-bet$^+$ B-cells are greatly expanded in *P.-falciparum*-infected patients

Because our previous studies in mice and previous reports in human malaria patients had shown a large increase in atypical memory B-cell (MBC) population upon infection with *Plasmodium* (*Patgaonkar et al., 2018*; *Pérez-Mazliah et al., 2018*; *Portugal et al., 2015*; *Rivera-Correa et al., 2017*; *Sullivan et al., 2016*; *Weiss et al., 2009*), we next analyzed the total levels of atypical MBCs in peripheral blood mononuclear cells (PBMC) in the cohort of *P.-falciparum*-infected returned travelers. We characterized atypical MBCs by the double expression of FcRL5 and T-bet, as both markers are highly expressed and characteristic of this population (*Figure 3A*). This population is known to be elevated in malaria patients from endemic areas (*Obeng-Adjei et al., 2017*; *Sullivan et al., 2016*). After gating out non-B-cells (CD19$^-$), we observed that FcRL5$^+$T-bet$^+$ B-cells are indeed expanded in the PBMC samples from *P.-falciparum*-infected German returned travelers when compared to samples from uninfected German controls (*Figure 3B*).

We further analyzed the cohort of *P.-falciparum*-infected returned travelers by considering two different groups: 1) tourists, who reported to be naïve to malaria, and 2) those visiting friends or relatives (VFR), who reported having at least one previous episode of malaria (*Table 1*). We did not observe any significant difference in the levels of atypical MBCs in the PBMC between these two groups (*Figure 3—figure supplement 1*). We also observed no significant gender difference between the levels of FcRL5$^+$T-bet$^+$ B-cells in PBMC (*Figure 3—figure supplement 2*). Furthermore, we found no significant correlation between the time after treatment at which samples were obtained (ranging from 0 to 31 days, *Table 1*) and the level of hemoglobin, which indicates that the variations in hemoglobin levels are not just a consequence of time after parasite clearance. We also observed a significant direct correlation between the levels of atypical MBCs and the days after treatment, which suggests that the levels of these cells continue to increase after treatment. This increase is compatible with the activation of atypical MBCs during infection and their continuing proliferation after parasite clearance (*Figure 3—figure supplement 3*).

Altogether, these initial results suggest an expansion of atypical FcRL5$^+$T-bet$^+$ B-cells in *P.-falciparum*-infected patients following acute infection.

## Atypical memory FcRL5$^+$T-bet$^+$ B-cells correlate with hemoglobin levels in *P.-falciparum*-infected returned travelers

We next sought to determine whether atypical MBCs correlate with hemoglobin levels in *P. falciparum* patients. For this purpose, we performed a B-cell sub-population gating analysis in PBMC samples from our cohort. Following classical gating strategies for all relevant B-cell (CD19$^+$) sub-populations from human PBMC (*Weiss et al., 2009*), we analyzed: (i) naïve B-cells (CD27$^-$CD21$^+$CD10$^-$), (ii) immature B-cells (CD10$^+$), (iii) plasma cells (CD27$^+$CD21$^-$CD20$^-$), (iv) classical MBCs (CD27$^+$CD21$^+$) and (v) atypical MBCs (FcRL5$^+$T-bet$^+$) (*Figure 4—figure supplement 1*). To define the atypical and classical MBC populations better, we analyzed the expression of T-bet in FcRL5$^+$ cells compared to classical MBCs, finding that the expression of T-bet is significantly higher in FcRL5$^+$ cells (*Figure 4—figure supplement 2*).

We found a significant inverse correlation between hemoglobin levels and atypical MBCs levels in the PBMC samples from *P. falciparum* patients (*Figure 4A*). Atypical MBCs did not significantly correlate with other relevant parameters such as parasitemia (*Figure 4—figure supplement 3*), but did correlate positively with the age of the patient (*Figure 4—figure supplement 4*), supporting previous studies that initially denominated these cells as Age-associated B-cells (*Phalke and Marrack, 2018*). We also analyzed a possible relation of atypical MBCs with thrombocytopenia, another complication that frequently accompanies malaria and has an autoimmune component to its pathology (*Lacerda et al., 2011*). We did not observe any significant correlation between the levels of atypical MBCs and thrombocyte counts in our cohort of patients (*Figure 4—figure supplement 5*), suggesting that atypical MBCs specifically correlate with anemia and not with other malaria-associated complications.

Interestingly, we observed that classical memory B-cells (CD27$^+$CD21$^+$) had a significant positive correlation with hemoglobin levels (*Figure 4B*). We did not find any significant correlation between hemoglobin level and percentage of naïve B-cells or immature B-cells, but observed a significant positive correlation with percentage of plasma cells (*Figure 4—figure supplement 6*).

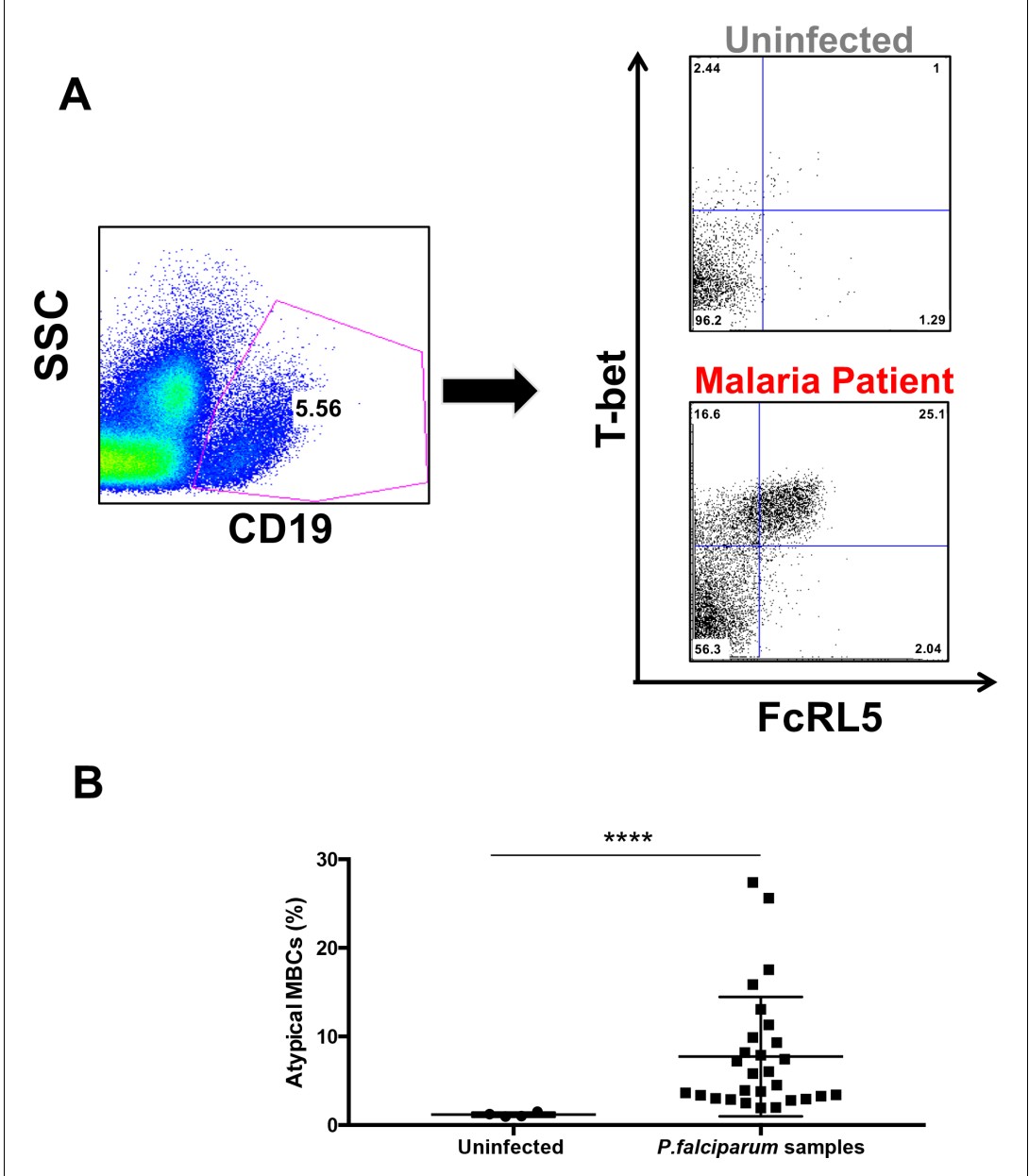

**Figure 3.** Atypical MBCs expand in *P.-falciparum*-infected patients and decline after treatment. (**A**) Gating strategy for the characterization of FcRL5[+] T-bet[+] B-cells (CD19[+]) with representative plots of one uninfected control and one *P. falciparum* patient. (**B**) Percentage of CD19[+] FcRL5[+] T-bet[+] B-cells in samples from uninfected controls and *P. falciparum* patients. Significance assessed by unpaired Student's t test. ****p≤0.0001.

DOI: https://doi.org/10.7554/eLife.48309.007

The following source data and figure supplements are available for figure 3:

**Source data 1.** Source data for *Figure 3*.
DOI: https://doi.org/10.7554/eLife.48309.014
**Figure supplement 1.** Atypical MBCs do not correlate significantly with patient background.
DOI: https://doi.org/10.7554/eLife.48309.008
**Figure supplement 1—source data 1.** Source data for *Figure 3—figure supplement 1*.
DOI: https://doi.org/10.7554/eLife.48309.009
**Figure supplement 2.** Atypical MBCs do not correlate significantly with patient gender.
DOI: https://doi.org/10.7554/eLife.48309.010
**Figure supplement 2—source data 1.** Source data for *Figure 3—figure supplement 2*.
DOI: https://doi.org/10.7554/eLife.48309.011

*Figure 3 continued on next page*

*Figure 3 continued*

**Figure supplement 3.** The time after treatment at which samples were collected correlates significantly with atypical MBCs but not with hemoglobin levels.
DOI: https://doi.org/10.7554/eLife.48309.012
**Figure supplement 3—source data 1.** Source data for *Figure 3—figure supplement 3*.
DOI: https://doi.org/10.7554/eLife.48309.013

Taken together, these results suggest that atypical MBCs, but not other B-cell subtypes, are specifically implicated in malaria-induced anemia in patients.

## Atypical memory B-cells correlate with anti-PS IgG antibodies in *P.-falciparum*-infected patients

During *Plasmodium* infections in mice, T-bet$^+$ B-cells secrete anti-PS IgG antibodies that induce premature clearance of uninfected erythrocytes, promoting malarial anemia (*Fernandez-Arias et al., 2016*; *Rivera-Correa et al., 2017*). The role of anti-PS IgG antibodies and the B-cells that secrete them during malarial anemia in *P.-falciparum*-infected patients has not been studied before. In this cohort of *P.-falciparum*-infected German returned travelers, we observed an inverse correlation between anti-PS IgG antibodies and hemoglobin, suggesting a role of these autoantibodies in promoting anemia in this cohort (*Figure 1B*). As both FcRL5$^+$T-bet$^+$ atypical B-cells and anti-PS IgG antibodies correlate with hemoglobin levels in our cohort, we assessed the relationship between the levels of FcRL5$^+$T-bet$^+$ atypical B-cells and anti-PS IgG antibodies. Our results show a significant positive correlation between FcRL5$^+$T-bet$^+$ atypical MBCs and anti-PS IgG antibodies (*Figure 5A*), possibly implicating these cells as the major producers of the antibodies. In accordance with the hemoglobin results (*Figure 4B*), we also found a significant inverse relationship between anti-PS IgG antibodies and classical MBCs (CD27$^+$CD21$^+$) (*Figure 5B*). As shown before for hemoglobin (*Figure 4—figure supplement 6*), neither naïve nor immature B-cells presented a significant correlation with the levels of plasma anti-PS IgG antibodies (*Figure 5—figure supplement 1*). Plasma cells, which previously correlated with hemoglobin levels, did not correlate with anti-PS IgG antibodies, suggesting that their role in anemia may be mediated through the secretion of antibodies that have different specificities. These results establish a relationship between atypical MBCs and anti-PS IgG antibodies.

As an inverse relationship between classical and FcRL5$^+$T-bet$^+$ atypical MBCs was found with hemoglobin and anti-PS antibodies, we analyzed whether there was any correlation between the levels of these two populations. Accordingly, we found a significant negative correlation between the levels of classical and FcRL5$^+$T-bet$^+$ atypical MBCs (*Figure 5C*). These results suggest a possible relationship between classical and atypical MBC populations, but may also be interpreted as the result of a robust proliferation of atypical MBCs, which would decrease the proportion of classical MBCs among the CD19$^+$ population even if their actual numbers had not decreased.

In addition, we assessed the levels of two other autoantibodies (anti-erythrocyte and anti-DNA) to further characterize the autoantibody repertoire that correlates with malaria anemia in *P.-falciparum*-infected patients. We first analyzed the relationship between anti-erythrocyte IgG antibodies and all of the B-cell sub populations. The analysis of memory B-cell subsets and anti-erythrocyte IgG antibodies resembles the anti-PS IgG antibodies correlations, showing a positive significant correlation between FcRL5$^+$T-bet$^+$ atypical MBCs. No significant correlation was found with classical memory, naïve or immature B-cell subsets. A positive correlation was observed with plasma cells, suggesting that other autoantibodies besides anti-PS are produced by these cells, possibly explaining the previous correlation of plasma cells with hemoglobin levels (*Figure 5—figure supplement 2*).

We also analyzed possible correlations of the levels of plasma anti-DNA antibodies with all of the B-cell subpopulations (*Figure 5—figure supplement 3*). We did not observe any significant correlation with atypical MBCs, suggesting that anti-DNA antibodies are not predominantly produced by this subpopulation of B-cells. The lack of correlation with other B-cell sub-populations (naïve, immature, plasma cell, and classical memory) does not provide any indications on the B-cell subtype that produces these antibodies. Anti-DNA antibodies do not correlate with hemoglobin levels in our

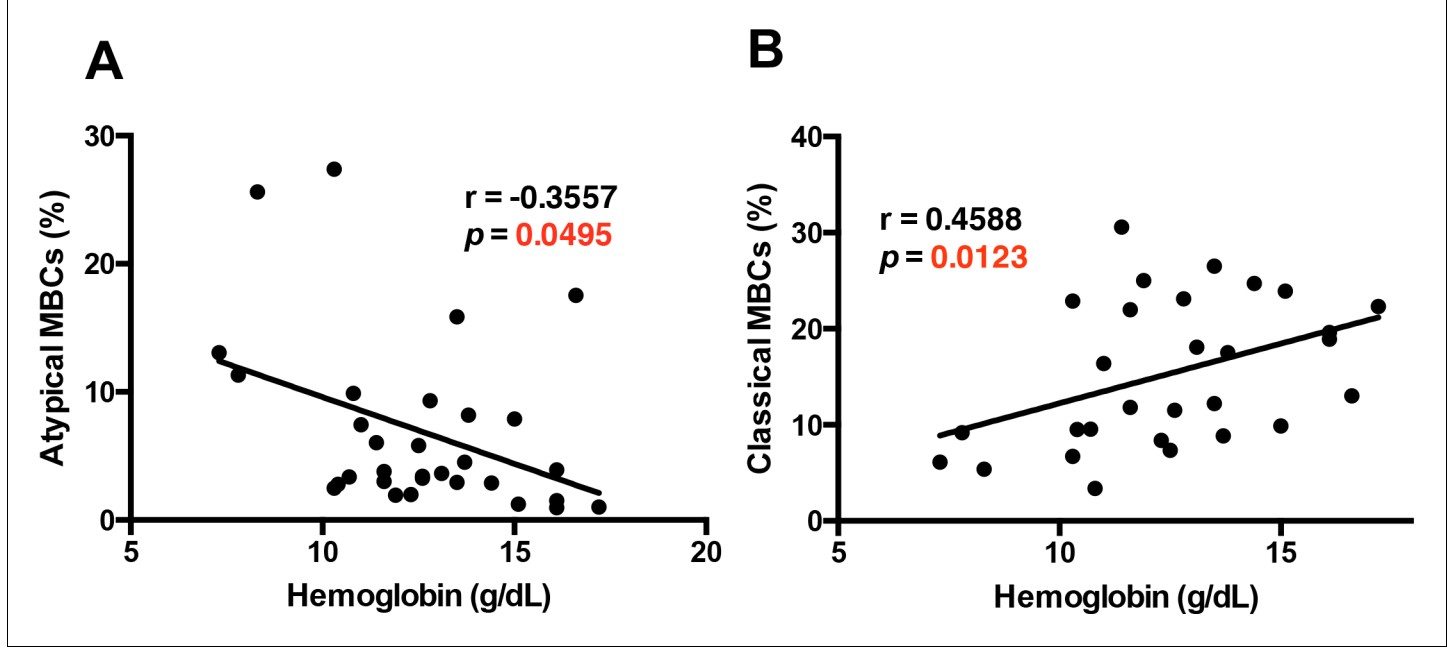

**Figure 4.** The atypical MBC subset correlates with the development of anemia in *P. falciparum* patients. Correlation analysis of atypical (**A**) and classical (**B**) MBC subsets from the PBMC of *P. falciparum* patients compared with hemoglobin levels. Significance was assessed by non-parametric Spearman correlation analysis.

DOI: https://doi.org/10.7554/eLife.48309.015

The following source data and figure supplements are available for figure 4:

**Source data 1.** Source data for *Figure 4*.

DOI: https://doi.org/10.7554/eLife.48309.027

**Figure supplement 1.** Gating strategy for relevant B-cell sub-populations.

DOI: https://doi.org/10.7554/eLife.48309.016

**Figure supplement 2.** Expression of T-bet in FcRL5[+] cells compared to classical MBCs.

DOI: https://doi.org/10.7554/eLife.48309.017

**Figure supplement 2—source data 1.** Source data for *Figure 4—figure supplement 2*.

DOI: https://doi.org/10.7554/eLife.48309.018

**Figure supplement 3.** Atypical MBCs do not correlate significantly with parasitemia.

DOI: https://doi.org/10.7554/eLife.48309.019

**Figure supplement 3—source data 1.** Source data for *Figure 4—figure supplement 3*.

DOI: https://doi.org/10.7554/eLife.48309.020

**Figure supplement 4.** Atypical MBCs correlate significantly with patient's age.

DOI: https://doi.org/10.7554/eLife.48309.021

**Figure supplement 4—source data 1.** Source data for *Figure 4—figure supplement 4*.

DOI: https://doi.org/10.7554/eLife.48309.022

**Figure supplement 5.** Atypical MBCs do not correlate significantly with thrombocyte levels.

DOI: https://doi.org/10.7554/eLife.48309.023

**Figure supplement 5—source data 1.** Source data for *Figure 4—figure supplement 5*.

DOI: https://doi.org/10.7554/eLife.48309.024

**Figure supplement 6.** Correlations of other B-cell subsets with hemoglobin levels in *P. falciparum* patients.

DOI: https://doi.org/10.7554/eLife.48309.025

**Figure supplement 6—source data 1.** Source data for *Figure 4—figure supplement 6*.

DOI: https://doi.org/10.7554/eLife.48309.026

cohort (*Figure 1E*), so it is likely that they do not play a role in this pathology and therefore are not expected to correlate with relevant B-cells subsets that contribute to anemia.

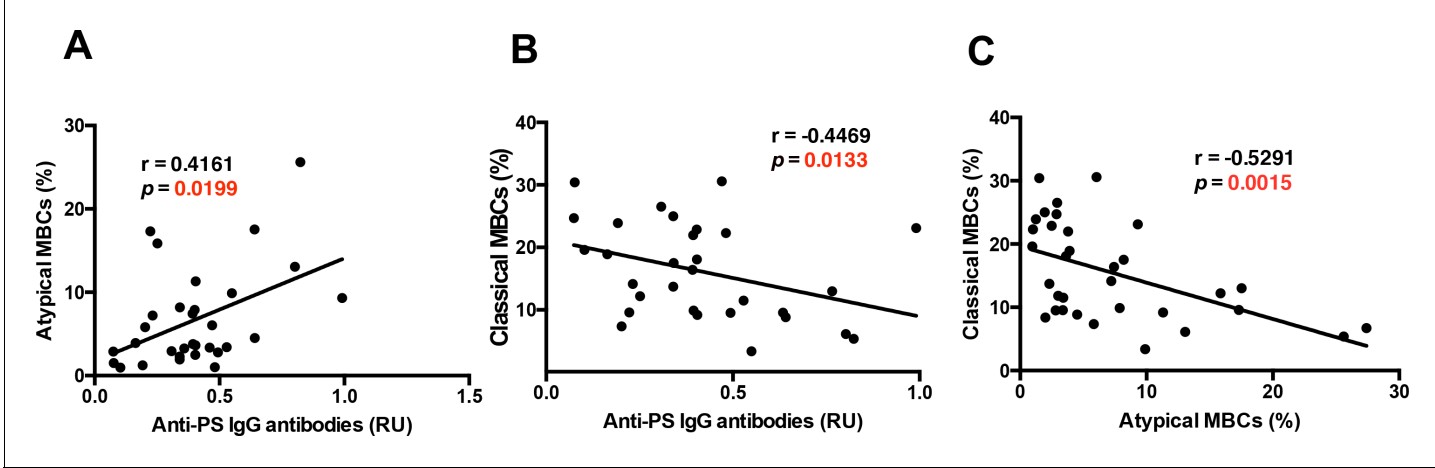

**Figure 5.** Anti-PS IgG antibodies show distinct correlations with classical and atypical MBC subsets in *P. falciparum* patients. Correlation analysis of levels of (A) atypical and classical (B) MBCs with anti-PS IgG antibody levels from the plasma of *P. falciparum* patients. (C) Correlation analysis of atypical and classical MBC levels. Significance was assessed by non-parametric Spearman correlation analysis.

DOI: https://doi.org/10.7554/eLife.48309.028

The following source data and figure supplements are available for figure 5:

**Source data 1.** Source data for *Figure 5*.
DOI: https://doi.org/10.7554/eLife.48309.035

**Figure supplement 1.** Anti-PS IgG antibodies do not correlate with other B-cell subsets in *P. falciparum* patients.
DOI: https://doi.org/10.7554/eLife.48309.029

**Figure supplement 1—source data 1.** Source data for *Figure 5—figure supplement 1*.
DOI: https://doi.org/10.7554/eLife.48309.030

**Figure supplement 2.** Correlations of anti-RBC IgG antibodies with B-cell subsets in *P. falciparum* patients.
DOI: https://doi.org/10.7554/eLife.48309.031

**Figure supplement 2—source data 1.** Source data for *Figure 5—figure supplement 2*.
DOI: https://doi.org/10.7554/eLife.48309.032

**Figure supplement 3.** Anti-DNA IgG antibodies do not correlate with the B-cell subsets analyzed in *P. falciparum* patients.
DOI: https://doi.org/10.7554/eLife.48309.033

**Figure supplement 3—source data 1.** Source data for *Figure 5—figure supplement 3*.
DOI: https://doi.org/10.7554/eLife.48309.034

## Atypical memory B-cells do not correlate with anti-PfEBA antibodies in *P.-falciparum*-infected patients

We also assessed the relationship between plasma anti-parasite antibodies (anti-PfEBA) and the different B-cell subsets in our cohort of *P.-falciparum*-infected patients. As specific plasma autoantibodies (anti-PS and anti-erythrocyte, but not anti-DNA) correlate distinctly with the atypical MBC subset and with anemia development, we questioned whether there is any correlation between any of the different B-cell subsets and anti-parasite antibodies (anti-PfEBA). This analysis showed no significant correlation between anti-PfEBA IgG antibodies and any of the B-cells subsets assessed (*Figure 6*). Anti-PfEBA IgG antibodies presented no significant correlation with anemia in patients (*Figure 1C*), so the lack of correlation with FcRL5[+]T-bet[+] atypical MBCs, which tightly correlate with anemia, is expected and further supports the lack of involvement of anti-parasite antibodies in malarial anemia.

## FcRL5[+]T-bet[+] atypical B-cells secrete anti-PS antibodies upon stimulation with *P.-falciparum*-infected erythrocytes in vitro

We have observed a strong significant correlation between FcRL5[+]T-bet[+] B-cells, anti-PS antibodies and malarial anemia in a cohort of *P.-falciparum*-infected patients (*Figures 3* and *4*). As these data from patient samples are limited to the analysis of correlations between different parameters, we aimed to determine directly whether activation of FcRL5[+]T-bet[+] atypical B-cells can induce the secretion of anti-PS antibodies. Expansion of T-bet[+] B-cells and secretion of anti-PS antibodies into

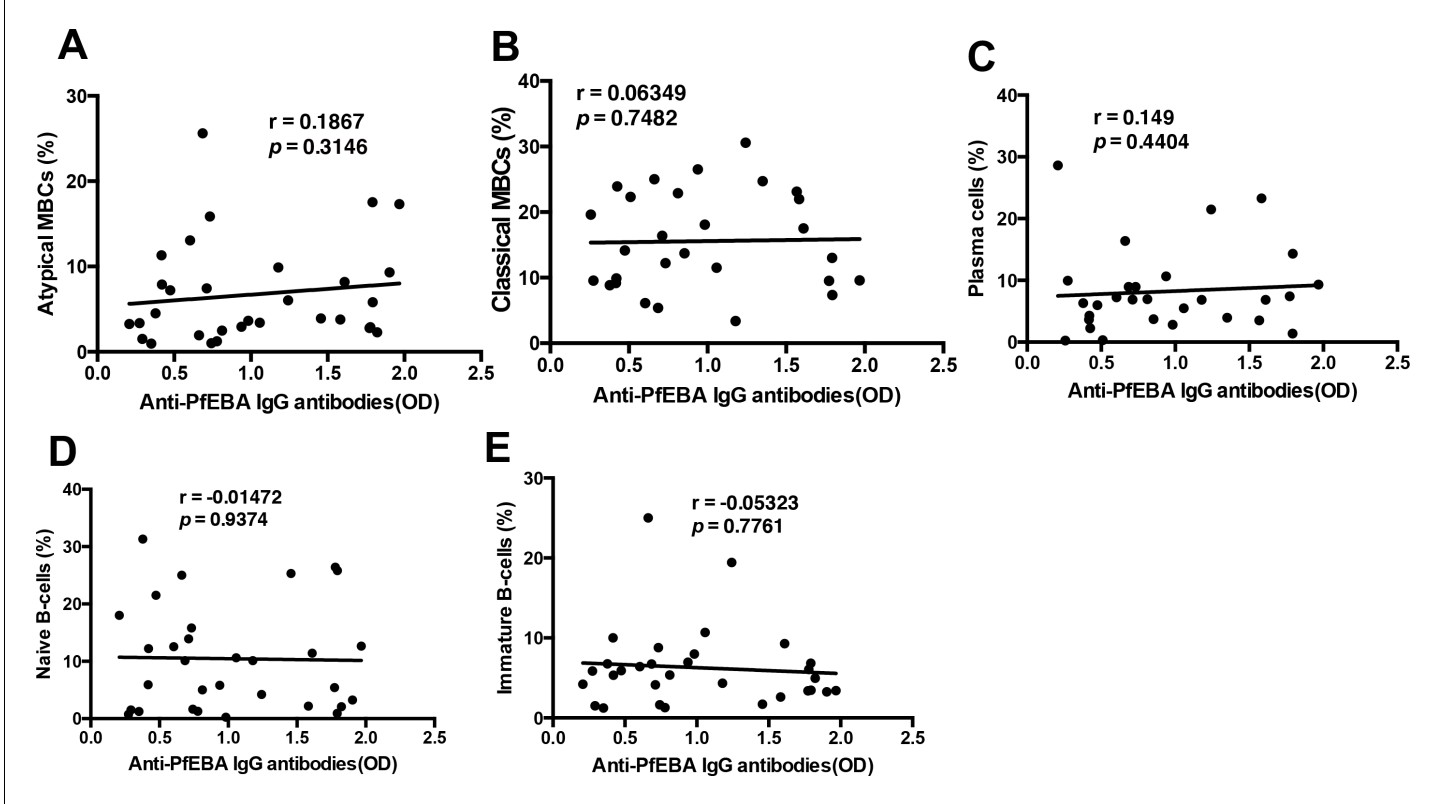

**Figure 6.** There is no significant correlation of anti-parasite *PfEBA* antibodies with relevant B-cell subsets from *P. falciparum patients*. Correlation analysis of (**A**) atypical MBCs, (**B**) classical MBCs, (**C**) plasma cells, (**D**) naïve B-cells, and (**E**) immature B-cells with anti-*P. falciparum* (PfEBA) IgG antibody levels from the plasma of *P. falciparum* patients. Significance was assessed by non-parametric Spearman correlation analysis.

DOI: https://doi.org/10.7554/eLife.48309.036

The following source data is available for figure 6:

**Source data 1.** Source data for *Figure 6*.

DOI: https://doi.org/10.7554/eLife.48309.037

the culture medium was observed in vitro after incubation of PBMC from healthy donors with lysates of *P.-falciparum*-infected erythrocytes (*Rivera-Correa et al., 2017*). To determine whether the T-bet[+] B-cells that are activated in these experiments are also FcRL5[+] and, more importantly, whether they specifically secrete anti-PS antibodies, we incubated PBMC from healthy US individuals with *P.-falciparum*-infected erythrocyte lysates. We observed a robust expansion of FcRL5[+]T-bet[+] B-cells compared to cells incubated with uninfected erythrocyte lysate or with no stimulation (*Figure 7A*). To investigate specifically whether these in vitro *P.-falciparum*-induced FcRL5[+] T-bet[+] atypical B-cells secrete anti-PS antibodies, we first enriched this population by selecting FcRL5[+] cells from PBMC stimulated with *P.-falciparum*-infected erythrocyte lysate. Remarkably, ELISPOT analysis of the antibody specificity of enriched FcRL5[+] cells showed significantly increased numbers of anti-PS-specific B-cells in the FcRL5[+] population when compared to enriched CD27[+] cells, which would represent predominantly classical memory B-cells and plasmablast/plasma cells (*Figure 7B*). No significant difference is observed between the number of FcRL5[+]antibody-secreting cells (ASCs) producing anti-PS and the number of anti-PfEBA cells, suggesting that the FcLR5[+] subset can efficiently produce both autoimmune and anti-parasite antibodies.

As FcRL5 is upregulated transiently on activated B-cells (*Dement-Brown et al., 2012*), the population expressing this molecule could include not only atypical B-cells but also any recently activated B-cell. However, we observed that CD27[+]-enriched cells had higher numbers of total ASCs than FcLR5[+] (*Figure 7—figure supplement 1*), which indicates that activated ASCs are found in both populations (CD27[+] and FcLR5[+]) and that the FcLR5[+] population does not include most of the activated B cells. Distinctly, quantification of PS-specific ASCs shows that these cells are more frequent

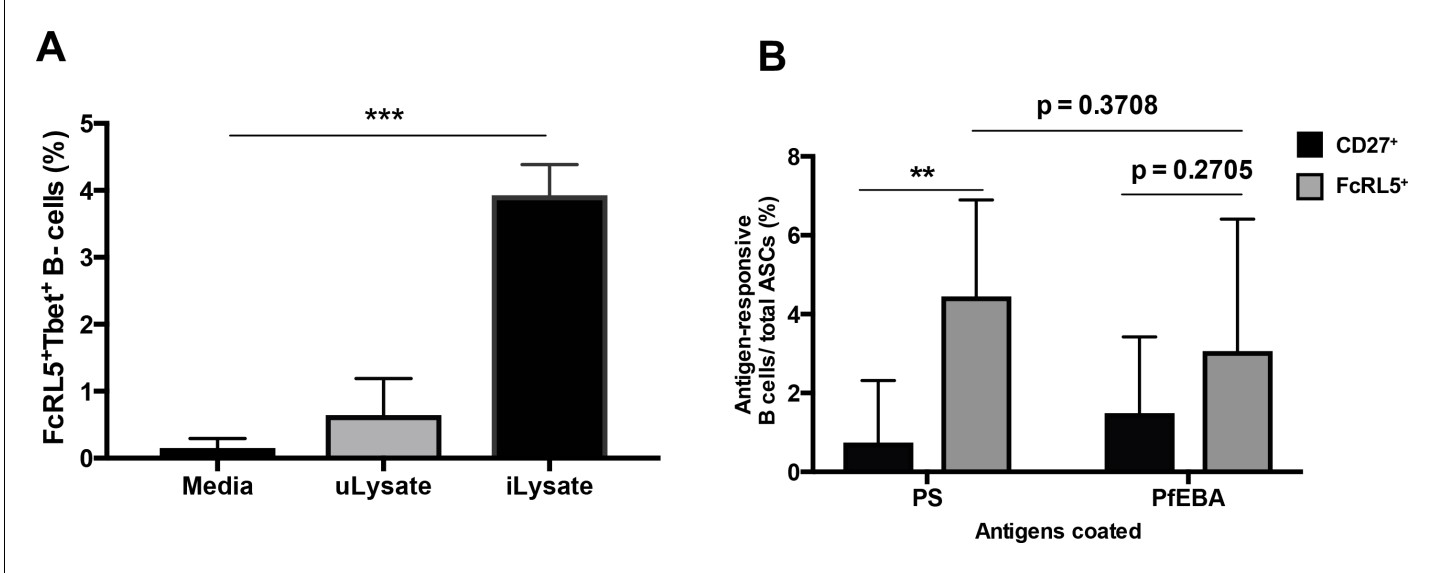

**Figure 7.** *P. falciparum* drives the expansion of human FcRL5[+] T-bet[+] B-cells that secrete anti-PS antibodies in vitro. (A) Percentage of T-bet[+]FcRL5[+] B-cells that expanded from the PBMCs of a healthy naïve donor after in-vitro exposure to either uninfected erythrocyte lysate (uLysate) or *P.* - *falciparum*-infected erythrocyte lysate (iLysate). (B) ELISPOT of enriched populations for either FcRL5 (gray bars) or CD27 (black bars) from PBMCs of healthy naïve US donors after in-vitro exposure to *P.-falciparum*-infected erythrocyte lysate (iLysate) (N = 3). ASC, antibody-secreting cells. Significance assessed by unpaired Student's t test. **p≤0.01, ***p≤0.001.

DOI: https://doi.org/10.7554/eLife.48309.038

The following source data and figure supplements are available for figure 7:

**Source data 1.** Source data for *Figure 7*.
DOI: https://doi.org/10.7554/eLife.48309.041

**Figure supplement 1.** Total antibody-secreting cells among the CD27[+-] and FcLR5[+]-enriched PBMC.
DOI: https://doi.org/10.7554/eLife.48309.039

**Figure supplement 2.** Stimulation with *P. falciparum* Histidine Rich Protein II (HRPII) does not stimulate the expansion of atypical MBCs in vitro.
DOI: https://doi.org/10.7554/eLife.48309.040

among FcLR5[+] cells than amongCD27[+] cells, despite having similar numbers of PfEBA-specific and total ASCs. These results indicate that both enriched populations have similar numbers of activated B-cells, but FcLR5[+] cells contain more anti-PS-secreting cells than CD27[+] cells.

Stimulation of PBMC with a specific *P. falciparum* antigen, Histidine Rich Protein II (HRPII), or with uninfected erythrocyte lysate did not induce a significant expansion of FcRL5[+]T-bet[+] cells in these cultures (*Figure 7—figure supplement 2*). This may be explained by the observation that the three different signals that are needed for optimal expansion of these cells in vitro—B-cell receptor stimulation, specific inflammatory cytokines (IFNγ/ IL−21) and *Plasmodium* DNA (*Phalke and Marrack, 2018*; *Rivera-Correa et al., 2017*)—are present in the co-cultures of PBMC with infected lysates, but not in co-cultures with uninfected lysates or purified proteins that lack *Plasmodium* DNA.

Taken together, these results indicate that atypical FcRL5[+]T-bet[+] B-cells produce anti-PS antibodies upon stimulation with parasite lysates in vitro and probably contribute to malarial anemia in patients through the secretion of anti-PS antibodies.

## Discussion

Malarial anemia in *P.-falciparum*-infected patients has been a complex subject of study because of its multi-factorial etiologies and the complicated logistics of endemic sites (*Lamikanra et al., 2007*; *Perkins et al., 2011*; *Price et al., 2001*). Although several studies in vitro and in mouse models have analyzed possible mechanisms mediating anemia in malaria (*Mourão et al., 2018*; *Mourão et al., 2016*; *Rivera-Correa et al., 2017*; *Safeukui et al., 2015*), very few studies have been able to translate their findings to malaria patients (*participants of the Hinxton retreat meeting on Animal*

*Models for Research on Severe Malaria et al., 2012*; *Lamikanra et al., 2007*; *Ndour et al., 2017*). Autoimmunity has been proposed to mediate malaria-associated anemia through the induced-clearance of uninfected erythrocytes bound to anti-PS antibodies (*Fernandez-Arias et al., 2016*). Studies in mice identified T-bet[+] B-cells as the immune cell type secreting these anti-PS antibodies (*Rivera-Correa et al., 2017*). In this study, we focused on *P. falciparum* patients to test whether malaria-associated anemia correlates with the autoimmune anti-PS response and with FcRL5[+]T-bet[+] B-cells, which would be consistent with a causal relationship.

The observed correlation of hemoglobin levels with anti-PS and anti-erythrocyte IgG levels, but not with anti-DNA IgG antibodies, confirms the specificity of the autoimmune response that contributes to malarial anemia in patients. However, this specificity may not be limited to PS, as the erythrocyte lysis capacity of the patient's plasma was only partially inhibited when anti-PS binding was blocked, suggesting that other antibody specificities may contribute to the lysis of erythrocytes (*Mourão et al., 2018*; *Mourão et al., 2016*). We also observed that anti-erythrocyte antibody levels correlate with hemoglobin levels, plasma cells and atypical MBCs frequency, and both B-cell subsets also correlate with hemoglobin levels. Taken together, these results suggest that other autoimmune antibodies in addition to anti-PS, possibly secreted by plasma cells and atypical MBCs, may contribute to anemia in malaria patients.

The direct correlation of anti-PS antibodies with levels of LDH [which are increased as a result of erythrocyte lysis during malarial anemia (*Sonani et al., 2013*; *White, 2018*)] and with the erythrocyte lysis capacity of the plasma supports the hypothesis that autoimmune anti-PS antibodies contribute to anemia in malaria through complement-mediated lysis. Previous results showed increased macrophage phagocytosis of uninfected erythrocytes from I-infected mice that was mediated by anti-PS antibodies (*Fernandez-Arias et al., 2016*). It is likely that both complement-mediated lysis and phagocytosis may be mediated by anti-PS antibodies, contributing to malaria-induced anemia.

T-bet[+] B-cells have been widely studied in the context of autoimmunity (*Myles et al., 2017*; *Rubtsova et al., 2015*; *Rubtsova et al., 2013*), in particular in relation to systemic lupus erythematosus (SLE) (*Phalke and Marrack, 2018*). In this context, these cells have been suggested as major candidates for drivers of the pathological anti-nuclear antibody response that mediates the SLE-associated pathologies (*Rubtsova et al., 2017*; *Wang et al., 2018*). In the context of malaria, T-bet was found to be a key marker of atypical MBCs that are expanded in individuals living in malaria endemic areas (*Obeng-Adjei et al., 2017*; *Portugal et al., 2017*). Here, we have also used FcRL5, in addition to T-bet, as a marker to define better atypical MBCs in *P. falciparum* patients (*Sullivan et al., 2015*), as T-bet alone can identify a rather heterogenic population (*Frasca et al., 2017*). Atypical MBCs have been characterized mainly after repeated seasonal infections (*Ayieko et al., 2013*; *Ndungu et al., 2013*; *Patgaonkar et al., 2018*; *Scholzen et al., 2014*; *Sullivan et al., 2015*; *Weiss et al., 2009*; *Weiss et al., 2010*), which is in agreement with our findings of increased levels of atypical MBCs in primary infections.

Some studies characterized ex-vivo atypical MBCs that had markedly reduced effector function and low antibody secretion capacity (*Obeng-Adjei et al., 2017*; *Portugal et al., 2017*; *Portugal et al., 2015*). Although the unresponsiveness of these cells to direct ex-vivo stimulation with different standard stimuli suggested that these cells were inactive (*Obeng-Adjei et al., 2017*), both atypical and classical MBCs from malaria patients are capable of secreting antibodies against blood-stage *P. falciparum* antigens (*Krishnamurty et al., 2016*; *Lugaajju et al., 2017*; *Muellenbeck et al., 2013*; *Portugal et al., 2017*) (*Kim et al., 2019*; *Pérez-Mazliah et al., 2018*; *Sundling et al., 2019*). This is in agreement with our results from PBMC stimulated in vitro, in which FcRL5-enriched cells show reactivity against both PS and the PfEBA antigen.

Atypical MBCs have also been described in *Plasmodium*-infected mice, where they were characterized as short-lived, and therefore not contributing to the long-lived anti malaria immune response (*Pérez-Mazliah et al., 2018*), but where they were able to limit the protective anti-parasite antibody response (*Guthmiller et al., 2017*). In our previous study in mice, we characterized T-bet[+] B-cells as the main producers of anti-PS IgG antibodies during malarial anemia (*Rivera-Correa et al., 2017*).

Although several studies have described atypical MBCs in individuals living in malaria endemic areas (*Kim et al., 2019*; *Muellenbeck et al., 2013*; *Obeng-Adjei et al., 2017*; *Portugal et al., 2015*; *Sullivan et al., 2015*; *Sundling et al., 2019*; *Weiss et al., 2009*), few studies had previously focused on these cells during the acute infection of individuals from non-endemic areas, where primary

infections can be observed. It is likely that atypical MBCs have frequently been characterized as exhausted memory B-cells secreting minimal antibodies because the studies were performed in individuals from endemic areas who had previously suffered numerous malaria infections. Our cohort offers a differential advantage in enabling the study of atypical MBCs during primary *Plasmodium* infections (in the group of tourists), allowing the characterization of these cells during their first response to malaria in a unique setting and thus dissecting their contribution in patients without other endemic site bystander infections. We also observed that the VFR group, whose members reported previous malaria infections, presents levels of atypical MBCs that are similar to those seen in first-infection patients. Although the time after their last malaria infection is variable in this group, these patients are very different from individuals suffering continuous reinfections in endemic areas.

The correlation of expansion of atypical MBCs with anemia in patients suggests that these cells have a role in malaria pathogenesis, providing clinical significance to our findings. In addition, the lack of correlation between the prevalence of these cells and other clinical parameters, such as parasitemia, gender, patient background and thrombocyte count, points to the specificity of the pathological effect.

In mice, T-bet$^+$ B-cells are major producers of anti-PS antibodies during malaria and promote anemia by inducing the clearance of uninfected erythrocytes that bind to these antibodies (*Fernandez-Arias et al., 2016*; *Rivera-Correa et al., 2017*). T-bet$^+$ B-cells in these mice are activated directly by *Plasmodium* DNA through TLR-9 alongside IFN-γ to produce anti-PS antibodies (*Rivera-Correa et al., 2017*). Our observations suggest that a similar mechanism contributes to human malarial anemia, as we observed specific direct correlations between the three key elements of this process: anti-PS antibodies, atypical MBCs and anemia. Our results in human samples also provide evidence that FcRL5$^+$T-bet$^+$ B-cells correlate with anemia. These cells are observed circulating in high levels in individuals living in malaria-endemic areas and present low antibody secretion ex vivo (*Portugal et al., 2015*). It is likely that at least some of these atypical MBCs may be derived from the T-bet$^+$FcRL5$^+$ B-cells observed in our study, which are generated during primary acute malaria but would lose their antibody-secreting capacity over time.

To complement the studies with patient samples, we took a direct approach to determine whether atypical MBCs can secrete anti-PS antibodies upon activation. The detection of anti-PS-secreting FcRL5$^+$ B-cells, but not of anti-PS-secreting CD27$^+$ B-cells, after stimulation with *P.-falciparum*-infected erythrocyte lysates, directly links FcRL5$^+$ B-cells with autoimmune antibody secretion in response to parasite stimulation.

Collectively, results from the analysis of patient samples and from in vitro stimulation of healthy PBMC, suggest that atypical T-bet$^+$FcRL5$^+$ memory B-cells contribute to malarial anemia in *P.-falciparum*-infected patients through anti-PS antibody secretion. They also strengthen the dichotomy within memory B-cells, where the atypical subset correlates with autoimmune anemia while the classical subset correlates with hematological health. As there were no significant correlations between anti-PS IgG antibodies and the other B-cell subsets (naïve, immature, classical memory or plasma cells), or of other antibodies (DNA, PfEBA) with anemia or with atypical memory B-cells, these results suggest specificity and support the hypothesis that atypical T-bet$^+$FcRL5$^+$ memory B-cells secrete anti-PS IgG antibodies that contribute to malarial anemia in *P.-falciparum*-infected patients.

In summary, our results provide the first mechanistic evidence of autoimmune-mediated malaria anemia in patients, and suggest that atypical T-bet$^+$FcRL5$^+$ B-cells are major promoters of this pathology in *P.-falciparum*-infections. Given the need for novel targeted treatments for malarial anemia, which still presents high prevalence and mortality, the unique phenotype and specificity of these cells secreting anti-PS antibodies could enable biomarker identification and the development of targeted therapeutics.

# Materials and methods

**Key resources table**

| Reagent type (species) or resource | Designation | Source or reference | Identifiers | Additional information |
|---|---|---|---|---|
| Biological sample (*Homo sapiens*) | CPD backed cells | Interstate Blood bank | | |
| Antibody | Anti-human CD20 (mouse monoclonal) | Biolegend | 302304 | 1:100 |
| Antibody | Anti- T-bet (mouse monoclonal) | Biolegend | 644810 | 1:100 |
| Antibody | Anti-human CD11c (mouse monoclonal) | Biolegend | 301604 | 1:100 |
| Antibody | Anti-human CD27 (mouse monoclonal) | Biolegend | 302806 | 1:100 |
| Antibody | Anti-human CD21 (mouse monoclonal) | Biolegend | 354910 (FITC) 354906 (APC) | 1:100 |
| Antibody | Anti-human FcRL5 (mouse monoclonal) | Biolegend | 340306 | 1:100 |
| Antibody | Anti-human CD10 (mouse monoclonal) | Biolegend | 312210 | 1:100 |
| Antibody | Anti-human CD19 (mouse monoclonal) | Biolegend | 30228 | 1:100 |
| Antibody | Anti-human IgM- HRP (goat polyclonal) | Millipore | AP114P | 1:2000 |
| Antibody | Anti-human IgG-HRP (goat polyclonal) | GE Healthcare | NA933 | 1:2000 |
| Antibody | Anti-human FcRL5-biotin (mouse monoclonal) | Miltenyi Biotec | 130-105-993 | 1:100 |
| Antibody | Anti-human IgM unlabeled (mouse monoclonal) | Biolegend | 314–502 | 15 µg/ml |
| Antibody | Anti-human IgM-biotin (mouse monoclonal) | EMD Millipore | 411543 | 1 µg/ml |
| Peptide, recombinant protein | *P. falciparum* Erythrocyte Binding Antigen | BEI Resources MR-4 | #MRA-1162 | 15 µg/ml |
| Commercial assay or kit | True-Nuclear Transcription Factor Buffer Set | Biolegend | 424401 | |
| Commercial assay or kit | MycoAlert Mycoplasma Detection Kit | Lonza | LT07-118 | |
| Commercial assay or kit | TMB substrate | BD Biosciences | 555214 | |
| Commercial assay or kit | CD27 Microbeads human | Miltenyi Biotec | 130-051-601 | |
| Chemical compound, drug | Ionomycin | Life technologies | I24222 | 2.5 µM |
| Chemical compound, drug | Ficoll-Paquee Plus | GE Life Sciences | 17144002 | |
| Software, algorithm | GraphPad PRISM | GraphPad PRISM | | |
| Other | Phosphatidylserine | Sigma-Aldrich | P7769 | 20 µg/ml |
| Other | Calf Thymus DNA | Sigma-Aldrich | D4522 | 10 µg/ml |
| Other | Stop buffer | Biolegend | 423001 | |
| Other | Annexin V | Biolegend | 640902 | 0.5 µM |

*Continued on next page*

*Continued*

| Reagent type (species) or resource | Designation | Source or reference | Identifiers | Additional information |
|---|---|---|---|---|
| Other | X- VIVO 15 media | Lonza | 04-418Q | |
| Other | Human AB healthy plasma | Sigma-Aldrich | H4522 | |

## Study design and sample collection

Patients were recruited at the University Medical Center Hamburg-Eppendorf. Inclusion criteria were age between 18 and 65 years, hemoglobin >8 g/dl and a diagnosis of *P. falciparum* malaria by microscopy. All individuals gave written informed consent. The study protocol was approved by the ethics committee of the Hamburg Medical Association (PV4539). Plasma and PBMC were isolated from peripheral venous blood by Ficoll purification and stored at −80℃ until temperature-controlled transportation to New York University. The sample size was limited by the number of German patients reporting with *P. falciparum* infection at University Medical Center Hamburg-Eppendorf during one year. We obtained samples from 24 patients (at two different times after infection for seven of them) and four uninfected controls (*Table 1*). Spearman analysis of the seven pairs of repeated samples showed no significant correlation in the levels of atypical MBCs between repeated measurements. All patients received anti-malaria treatment on the day of presentation, which is considered day 0. Patients were classified as tourists or VRF. The latter is not a homogeneous group as it encompasses people who were born in their (non-malaria endemic) country of residence as well as people who had arrived in the current country of residence at any time before their current infection.

## *P. falciparum* culture and isolation

Erythrocyte asexual stage cultures of the *P. falciparum* strain 3D7 were maintained at 5% hematocrit in RPMI 1640, 25 mM HEPES supplemented with 10 µg/ml gentamycin, 250 µM hypoxanthine, 25 mM sodium bicarbonate, and 0.5% Albumax II (pH 6.75) under atmospheric conditions of 5% oxygen, 5% carbon dioxide, and 90% nitrogen. Magnetic separation of late stages with MACS cell separation columns (Miltenyi Biotec) was used for culture synchronization and to isolate late-stage-infected erythrocytes for use in experiments. For experiments with lysates, late-stage infected erythrocytes were lysed by 10 freeze/thaw cycles. *P. falciparum* culture supernatants were tested for mycoplasma contamination using the MycoAlert Mycoplasma Detection Kit (Lonza LT07-118) and found to be negative.

## Human PBMC enrichment

Peripheral venous blood from healthy malaria-naïve donors was obtained on the day of the experiment at the New York University Clinical and Translational Science Institute with sodium citrate as anticoagulant. Institutional Review Board (IRB) approval was obtained at New York University School of Medicine. PBMC were enriched using Ficoll-Paque PLUS (GE Life Sciences). All recruited volunteers provided written informed consent prior to blood donation.

## Flow cytometry

All flow cytometry was performed on a FACSCalibur (Becton Dickinson, Franklin Lakes, NJ) and analyzed with FlowJo (Tree Star, Ashland, OR). All Abs for FACS were purchased from BioLegend (San Diego, CA). For PBMC assays, PBMC were stained with anti-human: FITC anti-CD20 (2H7), PE anti-T-bet (4B10), FITC anti-CD11c (3.9), FITC anti-CD27 (O323), FITC anti-CD21 (Bu32), APC anti-CD21 (Bu32), APC anti-FcRL5 (509f6), APC anti-CD10 (HI10a), and PRCP anti-CD19 (HIB19). Intracellular T-bet staining was performed using the True-Nuclear Transcription Factor Buffer Set (Biolegend) and following manufacturer's instructions. Two to three technical replicates (independent labeling of PBMC and FACs analysis) for B-cell subpopulations were performed when the number of PBMC collected from each patient allowed for it (15 samples). The average value of technical replicates for each sample was used for statistical analysis.

## ELISA

Costar 3590 ELISA plates were coated with PS at 20 µg/ml or human uninfected erythrocytes lysate ($10^9$ erythrocytes/ml in PBS) diluted 1:500 in 200 proof molecular biology ethanol or with *P. falciparum* Erythrocyte Binding Antigen (PfEBA, which was obtained through BEI Resources, MR4, NIAID, NIH) or Calf Thymus DNA (Sigma) at 10 µg/ml in PBS 1X, and allowed to evaporate (PS) at RT for >16 hr of incubation at 4°C. Plates were washed five times with PBS 0.05% Tween 20 and then blocked for 1 hr with PBS 3% BSA. Plasma from patients was diluted at 1:100 in blocking buffer and incubated for 2 hr at 37°C. Plates were washed again five times and incubated with anti-human IgG-HRP (GE Healthcare) for 1 hr at 37°C. Plates were washed five more times and TMB substrate (BD Biosciences) was added until the desired color was obtained. The reaction was stopped by with Stop buffer (Biolegend) and absorbance was read at 450 nm. The mean OD at 450 nm from triplicate wells was compared with the same dilution of a reference positive serum to calculate relative units (RU). For human PBMC ELISAs, a similar process was performed but using human erythrocyte lysates or PS for coating, undiluted PBMC culture supernatants and anti-human IgM-HRP (Millipore) for detection. Three technical replicates for each plasma sample (independent wells in the same plate) were performed for ELISA. The average value of technical replicates for each sample was used for linear regression analysis. Each ELISA was performed at least twice. Only one representative result is shown.

## Erythrocyte lysis

Assessment of the erythrocyte lysis capacity of plasma was performed following previously described methods with small modifications (*Meulenbroek et al., 2014*). First, fresh healthy donor erythrocytes were treated with ionomycin at 2.5 µM (Life Technologies) to stress erythrocytes and to induce exposure of PS (*Lang et al., 2006*). Erythrocytes were then washed twice with PBS and incubated with heat-inactivated plasma from either patients or uninfected controls (8% of total volume) along with 3.5% of AB type healthy plasma (Sigma) as a complement source for 1.5–2 hr. For Annexin V blocking experiments, the same protocol was used but 0.5 µM Annexin V (Biolegend) or its buffer alone were preincubated with the RBCs for 30 min. Plates were then spun down with reduced break and supernatants were carefully collected. Supernatants were read in a spectrophotometer at 414 nm to assess erythrocyte lysis. Results are shown as percentage of maximal lysis (erythrocytes lysed by water).

## ELISPOT assay

ELISPOTs were performed as previously reported (*Rivera-Correa et al., 2017*). PBMC from a healthy US donor were obtained. Human PBMC were seeded in flat 96-wells at a density of $2.5 \times 10^4$ per well. *P.-falciparum*-infected erythrocyte lysates were prepared as mentioned before and added at a ratio of 1:10 (PBMC:erythrocytes) and cultured in serum-free hematopoietic cell X- VIVO 15 medium (Lonza) for 6 days. Specific B-cell populations were enriched through magnetic bead sorting (Miltenyi) by positive selection with a combination of purified biotinylated anti-FcRL5 antibody (atypical)/anti biotin beads or anti-CD27 (plasma/classical memory cell) coated magnetic beads. Enrichment yield was assessed by flow cytometry prior to addition to plate.

For ELISPOT, $5 \times 10^4$ cells were added per well and incubated in X- VIVO 15 medium (Lonza) in 96-well Costar 3590 ELISA plates (Corning Life Sciences, Tewksbury, MA) precoated with either capture anti-IgM (15 µg/ml), PS (100 µg/ml in ethanol) (Sigma, St. Louis, MO), PfEBA (15 µg/ml) or PBS 10% BSA as control for 20 hr at 37°C with 5% $CO_2$. Following extensive washings, anti-human IgM biotinylated detection antibody (EMD Millipore) was added at 1 µg/ml diluted in PBS 0.5% FBS for 2 hr at RT. Streptavidin-horseradish peroxidase (Mabtech AB, Nacka Strand, Sweden) was added diluted in PBS 0.5% FBS for 1 hr at RT. Plates were developed with TMB substrate (Mabtech AB, Nacka Strand, Sweden) for 15–20 min and then washed extensively with water. Spots were quantified by microscopy. Spots in wells coated with PS or PfEBA are representative of antigen-responsive B-cells. Spots in wells coated with anti-IgM are representative of total antibody secreting cells (ASC).

Biological replicates (samples from three different healthy donors in three independent experiments) were used for the ELISPOT assay and the related FACs analysis. Three technical replicates (independent wells in the same plate) were performed for each experiment.

## Statistical analysis

Data were analyzed using Prism (GraphPad Software). Unpaired t-tests were used to identify statistical differences between groups of samples. A p-value of <0.05 was considered significant. Correlations were performed using non-parametric Spearman correlation analysis. Error bars represent the standard deviations (SD) of data from all of the patients or donors used in each experiment.

## Acknowledgements

This work was supported in part by the National Institutes of Health (NIH) institutional training grants 5T32AI100853-03 and 5T32AI007180 to JRC and by a German Center for Infection Research (DZIF) grant to TR (TI07.001_Rolling). Healthy donors blood draw was performed at NYU CTSI, the National Center for the Advancement of Translational Science (NCATS), NIH. We acknowledge Marisol Zuniga from the Rodriguez Lab at NYU for her help with human PBMC isolation and parasite culture. The content is solely the responsibility of the authors and does not necessarily represent the official views of the NIH.

## Additional information

### Funding

| Funder | Grant reference number | Author |
| --- | --- | --- |
| National Institute of Allergy and Infectious Diseases | 5T32AI100853 | Juan Rivera-Correa |
| National Institute of Allergy and Infectious Diseases | 5T32AI007180 | Juan Rivera-Correa |
| Deutsches Zentrum für Infektionsforschung | TI07.001_Rolling | Thierry Rolling |

The funders had no role in study design, data collection and interpretation, or the decision to submit the work for publication.

### Author contributions

Juan Rivera-Correa, Conceptualization, Data curation, Formal analysis, Investigation, Methodology, Writing—original draft; Maria Sophia Mackroth, Thomas Jacobs, Julian Schulze zur Wiesch, Thierry Rolling, Recruited patients, gathered consent, collected and shipped samples; Ana Rodriguez, Conceptualization, Data curation, Formal analysis, Supervision, Investigation, Methodology, Writing—original draft, Writing—review and editing

### Author ORCIDs

Juan Rivera-Correa https://orcid.org/0000-0003-4822-0005
Ana Rodriguez https://orcid.org/0000-0002-0060-3405

### Ethics

Human subjects: Patients were recruited at the University Medical Center Hamburg-Eppendorf. Inclusion criteria were age between 18 and 65 years, hemoglobin >8g/dl and a diagnosis of *P. falciparum malaria* by microscopy. All individuals gave written informed consent. Participant data was transmitted to the United States after double pseudonymization and without any protected health information. The study protocol was approved by the Ethics committee of the Hamburg Medical Association (PV4539).

### Decision letter and Author response

Decision letter https://doi.org/10.7554/eLife.48309.044
Author response https://doi.org/10.7554/eLife.48309.045

## Additional files

### Supplementary files

• Transparent reporting form DOI: https://doi.org/10.7554/eLife.48309.042

### Data availability

All data generated or analysed during this study are included in the manuscript and supporting files. Source data files have been provided for all figures.

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
