## [Decision Letter]

**Acceptance summary:**

Rivera-Correa and colleagues probe an important question concerning malaria and anemia. On the basis of results recorded from the "first-time" malaria-infected adult subjects, the authors construct correlations that anti-phosphatidylserine autoantibodies, secreted by *Plasmodium falciparum*-induced atypical memory B cells, are involved in the lysis of uninfected red blood cells, thus leading to anemia. This work, touching on the etiology of malaria-associated anemia, represents an important aspect of malaria-associated pathology. These clinical observations increase our understanding of the processes that potentially lead to anemia and thus provide a framework for both researchers and clinicians to develop effective approaches for treating malaria-induced pathologies.

**Decision letter after peer review:**

Thank you for sending your article entitled "Atypical memory B-cells are associated with *Plasmodium falciparum* anemia through anti-phosphatidylserine antibodies" for peer review at *eLife*. Your article is being evaluated by three peer reviewers, one of whom is a member of our Board of Reviewing Editors, and the evaluation is being overseen by Satyajit Rath as the Senior Editor.

Given the list of essential revisions, including new experiments, the editors and reviewers invite you to respond within the next two weeks with an action plan and timetable for the completion of the additional work. We plan to share your responses with the reviewers and then issue a binding recommendation.

There following are some of the major concerns expressed by the reviewers.

1) Additional in vitro experiments need to performed to demonstrate:

a) That indeed the T-bet^+^FcRL5^+^ B cells selectively do produce PS antibodies that correlate with anemia;

b) That the anti-PS antibodies do engage in the lysis of RBC.

c) That T-bet^+^ B cells correlate with anti-parasite antibody (responses against whole parasite lysates.

2) Regarding the correlative data presented in the manuscript, the concern is that the results were reported with linear regression. Given the small number of patients, several significant correlations may have been driven by 1 or 2 outliers. Therefore, nonparametric assessments would have been more appropriate. In addition, some re-analyses should be provided to account for repeated measurements within individuals.

3) To strengthen the authors' claim – if at all possible – it would be to conduct these experiments with samples from another, e.g., endemic cohort.

Reviewer #1:

In this manuscript Rivera-Correa and colleagues probe an important question concerning anti-PS-specific antibodies (autoimmune) produced by atypical B cells as a probable cause of anemia amongst persons suffering from malaria infection. On the basis of data obtained from a group of *Plasmodium falciparum*-infected subjects who returned from African countries, the authors constructed correlations about the increased frequency of atypical memory B cells expressing FcRL5 and producing anti phosphatidylserine (PS) antibodies that are ultimately involved in the lysis of uninfected red blood cells, hence anemia.

The authors have previously published similar work on malaria inducing anemia via the anti-PS antibodies in a mouse model and now they extend this investigation to human subjects, which is an important aspect of malaria-associated pathology and thus it adds significant value to the study. The authors also claim that their cohort with a "possibly" one time malaria infection enhances the importance of their correlative analyses as recurrent or chronic malaria infections amongst residents of endemic areas complicate the function of the atypical B cells, since for the most part these B cells are unresponsive.

Although the authors conducted their investigation with a seemingly adequate number of subjects, several key aspects of the study group such as malaria infection vis-à-vis sampling of PBMC are not well presented. Such omission makes it rather difficult to understand and properly interpret the results. For example, what level of certainty can the authors provide that this was only "one time malaria infection" and that the subjects did not suffer from malaria previously? In Table 1, the data show the days since treatment. Is it also the time when the PBMCs were sampled? If not, when and how many times were the PBMC drawn and sera obtained? According to the data provided in the Table, there is a rather high variation as to the time since the subjects received treatment. It is imperative to provide this explanation somewhere in the text, as the days vary considerably. i.e., day 1 – 31. Is it possible that because of these variations, the results presented in Figure 1A, could be interpreted as showing two separate groups? Although, one does expect a certain level of variation in the level of any lymphocyte subsets in human populations, do the results showing T-bet^+^ B cells levels in *P. falciparum* infected persons reflect this expected variation, or do they reflect the time of treatment, the level of parasitemia, etc? These issues do need to be addressed to properly interpret the results. Also, it is not clear what the percentages in Figure 2B indicate. Are these percentages of T-bet^+^ B cells based on the number of CD19^+^B cells? Please clarify.

The key observation in this study is that the atypical FcRL5^+^B cells correlate with the production of anti-PS IgG antibodies, which in turn correlates with anemia. I would presume that the sera from which these antibodies were derived are still available. Additional (in vitro) experiments should be considered to demonstrate that the anti-PS IgG containing sera or isolated anti-PS antibodies derived from the Pf infected subjects lyse red blood cells. Results from these experiments would solidly confirm the correlative data presented by the authors.

As concerns experiments shown in Figure 6B, it is not entirely clear how the experiments were done and how the percentage of ELISPOTs were calculated? Please explain and explain abbreviation for ASC.

Reviewer #2:

Summary: This is a well written manuscript that provides novel insights about malaria-induced anemia. In summary, the authors show that anemia (low hemoglobin) in patients with malaria is correlated with anti-PS antibodies and frequencies of FCRL5^+^ and T-bet B cells. Anemia does not correlate with parasitemia or levels of anti-parasite antibodies (anti-EBA), and is inversely correlated with classical MBC frequency. T-bet^+^ B cells expand in patients with malaria, then contract after drug treatment. FCRL5^+^ B cells from healthy donor PBMCs, when incubated with Pf erythrocyte lysate, will secrete anti-PS antibodies (to a significantly greater extent than CD27^+^ cells); both populations secrete roughly equal levels of anti-Pf antibodies.

Major points

1) The key contribution of this manuscript is providing a potential autoimmune mechanism for malaria-associated anemia, as well as a potential function of T-bet^+^ atypical MBCs (but see objections to their interpretations on this below).

2) There needs to be a better effort to incorporate recent developments in our understanding of what atypical / T-bet^+^ cells are. If atypical/T-bet^+^ cells are recently activated, optimally derived cells, then it follows that they are the main cells contributing to the circulating antibody pool for a recent infection (presumably after having to differentiate into antibody-secreting cells, which the study doesn't really mention). It makes sense that they're not going to have a lot of anti-DNA antibody because presumably this hasn't been elicited recently (at least is not the dominant recent stimulus).

What's more interesting about their data is that the T-bet^+^ cells correlate with anti-PS, but not anti-EBA (a parasite antigen). It would be nice to know if this correlation holds for anti-parasite antibodies in general (e.g., ELISA against whole parasite lysate). We know from published work in both human and mouse that atypical MBCs include cells with anti-Pf specificity (Muellenbeck et al., Krishnamurthy et al., Perez-Mazliah et al., Kim et al) and they also are enriched for anti-HIV antibodies in chronic HIV patients. So it doesn't seem to be universally true that T-bet^+^ atypicals are autoreactive B cells. But it may be the case that during malaria, the active Ab response is skewed toward PS rather than parasite antigens, and if so this could be a potentially very interesting immune evasion mechanism.

3) It's hard to know what to take home from Figure 6 (in vitro stim of healthy donor PBMCs with infected RBC lysate). First of all, FCRL5 and CD27 aren't mutually exclusive markers for atMBCs and cMBCs; a decent fraction (5-45%) of CD27^+^ cells also express FCRL5 (Sullivan 2015, Kim 2019). So it's not clear what populations are being sorted in this expt-there is probably some overlap between the two. Also, since FCRL5 appears to be upregulated transiently on activated B cells (e.g. Dement-Brown et al. 2012), it's possible that this population just includes most or all of the activated B cells (which would be consistent with the greater # of ASCs observed in the FCRL5^+^ compared to CD27^+^ wells).

4) Because of the authors' focus on the nature of the T-bet^+^ FCRL5^+^ cells as producers of autoreactive (anti-PS) antibodies but not anti-parasite antibodies, a key question is whether the FCRL5^+^ subset selectively produces anti-PS. No stats are included in Figure 6B to compare the number of FCRL5^+^ ASCs producing anti-PS versus anti-PfEBA (gray bars)- are these numbers significantly different? It might be informative to try stimulating PBMCs with another, unrelated antigen (or even look at additional self and Pf antigens with the same stimulus) and see if this pattern holds true (i.e. that most ASCs are found in the FCRL5^+^ pool). This would be consistent with recent literature demonstrating that FCRL5^+^ B cells are responsive, recently activated cells (Perez-Mazliah et al., Kim et al.) but would weaken the authors' interpretation of this subset as a uniquely autoreactive subset.

5) The authors don't find a correlation between anti-EBA antibodies and any B cell subsets assessed (Figure 5, subsection “Anti-parasite antibodies do not correlate with FcRL5^+^ T-bet^+^ atypical B-cells during malarial anemia in *P. falciparum*-infected patients”). But we know that some anti-parasite antibodies are made. So… where are those antibodies coming from? Is EBA a good representative Ag? Are multiple B cell subsets contributing those Ab levels? Is it all coming from plasma cells in bone marrow and cannot be measured from PBMCs? If so, then is the fact that most B cell subsets (including plasma cells) don't correlate with anti-PS antibodies meaningful? Maybe the antibodies come from plasma cells in BM, and in fact T-bet^+^ B cells are an indirect marker of recency since infection, or immune status; and these things might also correlate with anemia severity.

Reviewer #3:

In this paper by Rivera-Correa and colleagues, the authors report an intriguing association between anti-phosphotidyl serine antibodies, FcRL5^+^T-bet^+^B cells, and hemoglobin levels among patients presenting with symptomatic malaria. The authors further describe in vitro experiments detailing that FCRL5^+^ T-bet^+^ B cells expand after coculture of naïve PBMC with Pf lysate, and that these expanded cells appear to produce anti-PS antibodies, which strengthens the observations reported and suggest that malarial anemia may be mediated in part by auto-immune antibody production by this B cell subset. The manuscript is clearly written and easy to follow. My major concern with the data presented in this manuscript is that the main results (Figures 1-5) are mainly correlations, and that the correlation results were reported with linear regression. With the small patient numbers (n=19) several of the "significant" correlations may have been driven by 1 or 2 "outliers" with lower hemoglobin, for example. Given the small numbers, nonparametric assessments would have been more appropriate (i.e. Spearman's correlation coefficients as opposed to linear regression). Importantly, in many of the figures, there appears to be 27 data points, but in the Materials and methods, the authors state that there were 19 patients (7 with convalescent samples). Thus, one would surmise that the figures reflect 1 data point from 12 individuals, and two data points from 7 individuals. Assuming that these repeated measurements are correlated, the authors should have performed some analysis to account for repeated measures within individuals (e.g. generalized estimating equations or mixed effects linear regression) since autocorrelation could also be driving these associations.

One possible way to strengthen the authors' claims would be to obtain and analyze samples from a secondary cohort (e.g., an endemic cohort, if obtaining samples from a traveler's clinic is too time consuming.) The authors defend the choice of using a traveler's cohort in the discussion, since in many instances this might represent a primary infection, although I think it would also be very interesting to see if frequencies of FCRL5^+^T-bet^+^ B cells correlate with anti-PS antibodies in endemic settings as well.

Materials and methods: 19 patients and 4 controls

Results section – 27 samples? This is misleading. Would state X patients, 27 unique samples. (but isn't it actually 20 patients? Your numbering is from 100-119, so you would also count patient 0?)

Concern in Figure 1A: 2 patients with Hb <5 but not in table? What are these 2 measurements? Assuming this reflects Hb measured in all patients at the time that parasitemia was assessed (?Day 0) This data should be included in Table 1.

Table 1 generally is difficult to follow and should be clarified (potentially in legend or with updated column headers)

– Days since treatment start – needs clarification that this is the day of PBMC/plasma/Hb sampling.

– Also – for 7 individuals, the 2nd visit is also detailed in the table. I would argue that this information should be on the same "row" in columns to the right rather than a separate row.

– Hb measurement is at the date of sampling – do you also have hemoglobin at date parasitemia measured? (as above)

– Why does parasite count in Table 1 oscillate between% and parasites/μl? Not easy to interpret this in the table. Presumably these were all converted to parasites/µl for use in Figure 1A. Would use one standard metric here.

Results section the authors state that hemoglobin does not correlate with parasitemia, but this is an incorrect statement – they are not "significantly" correlated, although, visually, a positive correlation is suggested.

Results section: – "expression of T-bet in CD27^-^CD21^-^FcRL5^+^ B-cells is directly correlated". Assuming that you assessed co-expression of T-bet in FCRL5^+^ cells? Would be a more direct analysis vs. showing the correlation.

Discussion

Paragraph one: you didn't actually test the hypothesis that malaria-associated anemia is mediated by an autoimmune anti-PS response – you tested the correlation per se, which would be consistent with a causal relationship. Would be a little more judicious in your discussion/conclusions.

[Editors' note: further revisions were requested prior to acceptance, as described below.]

Thank you for resubmitting your work entitled "Atypical memory B-cells are associated with *Plasmodium falciparum* anemia through anti-phosphatidylserine antibodies" for further consideration at *eLife*. Your revised article has been favorably evaluated by Satyajit Rath (Senior Editor) and two reviewers, one of whom is a member of our Board of Reviewing Editors.

In the revised version of the manuscript, Juan Rivera-Correa and colleagues describe an interesting observations concerning the cause of anemia that accompanies malaria in *Plasmodium falciparum* exposed persons. It appears that *Plasmodium falciparum* infected red blood cells activate B cells that express phenotypic markers indicative of atypical B cell population (FcRL5^+^T-bet^+^) and such induced B cells produce antibodies that are autoreactive against phosphatidylserine that cause lysis of uninfected red blood cells. The authors found that the population of these atypical B cells did increase in European travels, who suffered from malaria episode(s) while visiting African countries. The authors demonstrate that atypical FcRL5^+^T-bet^+^ B cells are greatly expanded in acute malaria in *P. falciparum*-infected patients and correlate with both anemia and plasma anti-PS antibody levels in these patients. The authors have also confirmed these observations in vitro conducted experiments. These are novel observations showing that human atypical B cells produce anti-PS autoantibodies that play a significant role in the pathogenesis of human malaria anemia.

The manuscript has been improved but a few remaining issues, mainly editorial in nature, need to be addressed before acceptance. The issues to consider for revision are outlined below:

1) The introduction to each section fails to capture and integrate narratively the flow of the results. Please make a few editorial edits to better highlight this flow and integration. For example, the first sentence of the second section of the Results reads "Our first aim was..". However, the first section of the Results section seems to be the first aim, which is to establish that autoantibodies correlate with anemia and erythrocyte lysis in malaria patients. Similarly, the first sentence of the third section reads, "Our main goal".. Please reword to simply state, "We next sought to determine whether B cell subsets described above correlate with hemoglobin levels.

2) Gating plots and analysis of Figure 3 and Figure 4. AS written, it seems as though the analysis reported in section 2 (Figure 3) and section 3 (Figure 4) use different gating strategies to define populations of atypical memory B cells. Hopefully this is not the case. For example, in section 2, it states that AtMBC were defined as CD19^+^, FCRL5^+^, T-bet^+^. However, in Section 3, it states that AtMBC were defined as CD19^+^CD21^-^CD27^-^FCRL5^+^T-bet^+^. Please 1) integrate the two "definitions" into one and present these in section 2 of the results, with one integrated gating plot included (not separated into one in Figure 3 and one in Figure 4 —figure supplement.

Relatedly- gating figure (Figure 4—figure supplement). Not a very convincing plot of atypical memory B cells as cd27^-^cd21^-^ cells (2.9%?). Do you have a more representative gating figure that can be included instead? (especially if your median in tourists is close to 9%., and given the beautiful plot shown in Figure 3?).

3) Results subsection “Anti-parasite antibodies do not correlate with FcRL5^+^ T-bet^+^ atypical memory B-cells in *P. falciparum*-infected patients” header. Response to only one parasite antigen was measured, so this section header should be changed to something similar to the Figure 6 figure legend.

One suggestion:

"Atypical memory B-cell frequencies do not significantly correlate with anti-PfEBA antibodies in *P. falciparum* infected patients".

4) In Discussion, paragraph seven, there are a duplication in text the correct text needs to be included.

---

## [Author Response]

There following are some of the major concerns expressed by the reviewers.1) Additional in vitro experiments need to performed to demonstrate:a) That indeed the T-bet^+^FcRL5^+^ B cells selectively do produce PS antibodies that correlate with anemia;b) That the anti-PS antibodies do engage in the lysis of RBC.c) That T-bet^+^ B cells correlate with anti-parasite antibody (responses against whole parasite lysates.2) Regarding the correlative data presented in the manuscript, the concern is that the results were reported with linear regression. Given the small number of patients, several significant correlations may have been driven by 1 or 2 outliers. Therefore, nonparametric assessments would have been more appropriate. In addition, some re-analyses should be provided to account for repeated measurements within individuals.3) To strengthen the authors' claim – if at all possible – it would be to conduct these experiments with samples from a another, e.g., endemic cohort.Reviewer #1:In this manuscript Rivera-Correa and colleagues probe an important question concerning anti-PS-specific antibodies (autoimmune) produced by atypical B cells as a probable cause of anemia amongst persons suffering from malaria infection. On the basis of data obtained from a group of Plasmodium falciparum-infected subjects who returned from African countries, the authors constructed correlations about the increased frequency of atypical memory B cells expressing FcRL5 and producing anti phosphatidylserine (PS) antibodies that are ultimately involved in the lysis of uninfected red blood cells, hence anemia.The authors have previously published similar work on malaria inducing anemia via the anti-PS antibodies in a mouse model and now they extend this investigation to human subjects, which is an important aspect of malaria-associated pathology and thus it adds significant value to the study. The authors also claim that their cohort with a "possibly" one time malaria infection enhances the importance of their correlative analyses as recurrent or chronic malaria infections amongst residents of endemic areas complicate the function of the atypical B cells, since for the most part these B cells are unresponsive.Although the authors conducted their investigation with a seemingly adequate number of subjects, several key aspects of the study group such as malaria infection vis-à-vis sampling of PBMC are not well presented. Such omission makes it rather difficult to understand and properly interpret the results. For example, what level of certainty can the authors provide that this was only "one time malaria infection" and that the subjects did not suffer from malaria previously?

We thank the reviewer for the insightful comments and detailed review. We have analyzed the patient’s data on previous malaria episodes finding that all patients in the Tourist group reported never suffering from malaria before, which is likely to be accurate since they were all born in Germany. On the other hand, all patients in the VFR group reported having at least one previous episode of malaria. The time from the last malaria episode was not reported. This information is now included in Results section.

In Table 1, the data show the days since treatment. Is it also the time when the PBMCs were sampled? If not, when and how many times were the PBMC drawn and sera obtained?

Table 1 has been modified to clarify that it shows the “Days since treatment start to sampling”. Each line represents a collected sample, when more than one sample was collected from the same patient two lines with the same patient ID are shown in the table.

According to the data provided in the Table, there is a rather high variation as to the time since the subjects received treatment. It is imperative to provide this explanation somewhere in the text, as the days vary considerably. i.e., day 1 – 31. Is it possible that because of these variations, the results presented in Figure 1A, could be interpreted as showing two separate groups? Although, one does expect a certain level of variation in the level of any lymphocyte subsets in human populations, do the results showing T-bet^+^B cells levels in P. falciparum infected persons reflect this expected variation, or do they reflect the time of treatment, the level of parasitemia, etc? These issues do need to be addressed to properly interpret the results.

To address this concern, we have performed the analysis of ‘days after treatment’ and hemoglobin, which showed no significant correlation, indicating that the variations on hemoglobin levels are not just a consequence of time after parasite clearance.

We also observed a significant direct correlation between the levels of atypical MBCs and the days after treatment, which suggests that the levels of aMBCs keep increasing after treatment. This increase in the levels of aMBCs with time after treatment is compatible with aMBCs being activated during infection and continuing proliferation after parasite clearance.

This is now included in Results section and in (Figure 2—figure supplement 3).

We have also clarified that all patients were treated at the day of presentation which is considered day 0 (now included in Materials and methods section).

Also, it is not clear what the percentages in Figure 2B indicate. Are these percentages of T-bet^+^B cells based on the number of CD19^+^B cells? Please clarify.

The percentages of T-bet^+^ B cells are based on the gating of CD19^+^ cells as indicated in Figure 2A. This is clarified in Figure legend 2.

The key observation in this study is that the atypical FcRL5^+^B cells correlate with the production of anti-PS IgG antibodies, which in turn correlates with anemia. I would presume that the sera from which these antibodies were derived are still available. Additional (in vitro) experiments should be considered to demonstrate that the anti-PS IgG containing sera or isolated anti-PS antibodies derived from the Pf infected subjects lyse red blood cells. Results from these experiments would solidly confirm the correlative data presented by the authors.

As suggested by the reviewer, we have tested the capacity of the patient’s plasma to lyse erythrocytes finding increased erythrocyte lysis by patient’s plasma compared to healthy controls and a direct correlation of levels of anti-PS antibodies and erythrocyte lysis capacity of plasma. The erythrocyte lysis capacity is partially inhibited by annexin V, which specifically binds to PS, and inhibits the binding of anti-PS antibodies. This experiment was performed with the samples from 6 patients and 3 healthy controls from which there was enough volume remaining. We have also observed that levels of anti-PS antibodies correlate with the levels of LDH in patients. This is now included in Results section and in new Figure 2 and commented in the Discussion.

As concerns experiments shown in Figure 6B, it is not entirely clear how the experiments were done and how the percentage of ELISPOTs were calculated? Please explain and explain abbreviation for ASC.

Materials and methods section includes now a detailed explanation on how the percentages were calculated in the ELISPOTs and the explanation for ASC (antibody-secreting cells).

Reviewer #2:Summary: This is a well written manuscript that provides novel insights about malaria-induced anemia. In summary, the authors show that anemia (low hemoglobin) in patients with malaria is correlated with anti-PS antibodies and frequencies of FCRL5^+^ and T-bet B cells. Anemia does not correlate with parasitemia or levels of anti-parasite antibodies (anti-EBA), and is inversely correlated with classical MBC frequency. T-bet^+^ B cells expand in patients with malaria, then contract after drug treatment. FCRL5^+^ B cells from healthy donor PBMCs, when incubated with Pf erythrocyte lysate, will secrete anti-PS antibodies (to a significantly greater extent than CD27^+^ cells); both populations secrete roughly equal levels of anti-Pf antibodies.Major points1) The key contribution of this manuscript is providing a potential autoimmune mechanism for malaria-associated anemia, as well as a potential function of Tbet+ atypical MBCs (but see objections to their interpretations on this below).2) There needs to be a better effort to incorporate recent developments in our understanding of what atypical / T-bet^+^ cells are. If atypical/T-bet^+^ cells are recently activated, optimally derived cells, then it follows that they are the main cells contributing to the circulating antibody pool for a recent infection (presumably after having to differentiate into antibody-secreting cells, which the study doesn't really mention). It makes sense that they're not going to have a lot of anti-DNA antibody because presumably this hasn't been elicited recently (at least is not the dominant recent stimulus).

We thank the reviewer for all insightful comments that have improved significantly the quality of the manuscript.

The recent literature in atypical / T-bet^+^ B-cells has provided insights of the heterogenic population that the T-bet marker alone can indicate. Double expression of T-bet and FcRL5 was used in our study to better define the population of atypical memory B-cells, as has been described in malaria patients (Sullivan et al. 2015). This point has been explained in the Discussion.

What's more interesting about their data is that the T-bet^+^ cells correlate with anti-PS, but not anti-EBA (a parasite antigen). It would be nice to know if this correlation holds for anti-parasite antibodies in general (e.g., ELISA against whole parasite lysate).

We agree with the reviewer that it would be interesting to see the correlation with whole anti-parasite antibodies, but we would predict that testing for this correlation with whole parasite lysates (either using lysates of infected RBCs or lysates of purified merozoites), will not be conclusive since the parasite (and infected RBC) lysates contain large amounts of PS.

We know from published work in both human and mouse that atypical MBCs include cells with anti-Pf specificity (Muellenbeck et al., Krishnamurthy et al., Perez-Mazliah et al., Kim et al) and they also are enriched for anti-HIV antibodies in chronic HIV patients. So it doesn't seem to be universally true that T-bet^+^ atypicals are autoreactive B cells. But it may be the case that during malaria, the active Ab response is skewed toward PS rather than parasite antigens, and if so this could be a potentially very interesting immune evasion mechanism.

We have included a sentence and the references in the Discussion section to clarify that atypical MBCs include cells with specificity against *P. falciparum* antigens, as previously described. This is in agreement with our results in Figure 6, where a population of FcRL5-enriched cells shows reactivity against PfEBA antigen.

3) It's hard to know what to take home from Figure 6 (in vitro stim of healthy donor PBMCs with infected RBC lysate). First of all, FCRL5 and CD27 aren't mutually exclusive markers for atMBCs and cMBCs; a decent fraction (5-45%) of CD27^+^ cells also express FCRL5 (Sullivan 2015, Kim 2019). So it's not clear what populations are being sorted in this expt-there is probably some overlap between the two. Also, since FCRL5 appears to be upregulated transiently on activated B cells (e.g. Dement-Brown et al. 2012), it's possible that this population just includes most or all of the activated B cells (which would be consistent with the greater # of ASCs observed in the FCRL5^+^ compared to CD27^+^ wells).

The main purpose of Figure 6 is to link directly atypical FcRL5^+^ B cells with the secretion of autoimmune anti-PS antibodies. For this purpose, we compared FcRL5^+^ cells and CD27^+^ cells, which will differentially include plasmablasts/plasma cells as well as classical Memory B-cells. As the reviewer points out, these markers are not mutually exclusive and both enriched populations will include recently activated cells. In fact, CD27^+^ enriched cells had higher numbers of total antibody secreting cells (ASCs) than FcRL5^+^ when analyzed by total IgM spots (now included as Figure 7—figure supplement 1), which indicates that activated ASCs are found in both populations (CD27^+^ and FcRL5^+^) and that FcRL5^+^ population does not include most or all of the activated B cells.

Distinctly, quantification of PS-specific ASCs show that these cells are more frequent among FcRL5^+^ cells compared to CD27^+^ cells, despite having similar number of PfEBA-specific ASCs. These results indicate that even if CD27^+^ cells have a higher proportion of ASCs in general, FcRL5^+^ cells contain more anti-PS secreting cells. A sentence explaining these results is now included in Results section.

4) Because of the authors' focus on the nature of the T-bet^+^ FCRL5^+^ cells as producers of autoreactive (anti-PS) antibodies but not anti-parasite antibodies, a key question is whether the FCRL5^+^ subset selectively produces anti-PS. No stats are included in Figure 6B to compare the number of FCRL5^+^ ASCs producing anti-PS versus anti-PfEBA (gray bars)- are these numbers significantly different?

No significant difference is observed between the number of FCRL5^+^ ASCs producing anti-PS versus anti-PfEBA (now included in the graph). This suggests that the FcLR5 subset can efficiently produce both autoimmune and anti-parasite antibodies. This is now explained in the Results section.

It might be informative to try stimulating PBMCs with another, unrelated antigen (or even look at additional self and Pf antigens with the same stimulus) and see if this pattern holds true (i.e. that most ASCs are found in the FCRL5^+^ pool). This would be consistent with recent literature demonstrating that FCRL5^+^ B cells are responsive, recently activated cells (Perez-Mazliah et al., Kim et al.) but would weaken the authors' interpretation of this subset as a uniquely autoreactive subset.

As requested by the reviewer, we have attempted the stimulation of PBMC with a specific *P. falciparum* antigen, HRPII, and with uninfected erythrocyte lysate. We did not observe a significant expansion of T-bet^+^ FCRL5^+^ cells in these cultures (now included as Figure 7—figure supplement 2). This may be explained by the observation that three different signals are needed for optimal expansion of these cells in vitro: B-cell stimulation, specific inflammatory cytokines (IFNγ) and Plasmodium DNA (Rivera-Correa et al. 2017 Nat. Comm.) These three stimuli are present when PBMCs are incubated with whole *P. falciparum*-infected erythrocyte lysates, hence providing all the signals needed for expansion of T-bet^+^ FCRL5^+^ and production of anti-PS antibodies. However, no parasite DNA is present when the stimulus is a purified protein or uninfected erythrocyte lysates. This is now explained in the Results section.

5) The authors don't find a correlation between anti-EBA antibodies and any B cell subsets assessed (Figure 5, subsection “Anti-parasite antibodies do not correlate with FCRL5^+^ T-bet^+^ atypical B-cells during malarial anemia in P. falciparum-infected patients”). But we know that some anti-parasite antibodies are made. So… where are those antibodies coming from? Is EBA a good representative Ag? Are multiple B cell subsets contributing those Ab levels?

As the reviewer indicates, anti-Plasmodium antibodies are produced by at least two different subsets of B cells: atypical and classical (Muellenbeck et al., 2013), which may explain the lack of significant correlation with any of them in our study, which has a limited number of samples, and therefore is only able to detect strong correlations.

The differential correlation of anti-PS antibodies with atypical B cells versus other B cell subsets should be interpreted as these other subsets may contribute to a lesser extent (or do not contribute at all) to the production of anti-PS antibodies when compared to atypical B cells. We have worded the conclusions carefully to avoid over-interpretation of the results.

Is it all coming from plasma cells in bone marrow and cannot be measured from PBMCs? If so, then is the fact that most B cell subsets (including plasma cells) don't correlate with anti-PS antibodies meaningful? Maybe the antibodies come from plasma cells in BM, and in fact T-bet^+^ B cells are an indirect marker of recency since infection, or immune status; and these things might also correlate with anemia severity.

We would like to point out that, although atypical MBCs increase with time after treatment, suggesting that they could be a marker of recency, hemoglobin levels do not correlate with time after treatment (Figure 3—figure supplement 3), indicating that the time at which samples were collected is not the cause of the correlation between atypical MBC and hemoglobin levels.

Previous studies have established that malaria patients present antibody-secreting classical and atypical MBCs in the circulation (Muellenbeck et al., 2013), indicating that at least a fraction of the total antibodies are not produced by plasma cells in the bone marrow. Bone-marrow plasma cells are normally long-lived and produce low levels of high affinity antibodies against foreign antigens, but they are not considered producers of autoimmune antibodies in healthy adults (Lightman et al. 2019). Additionally, data from Plasmodium infections in mice (Fernandez-Arias et al., 2016 and Rivera-Correa et al. 2017) shows that anti-PS antibody levels decrease rapidly after infection, which would be unusual for bone marrow resident plasma cells.

Reviewer #3:In this paper by Rivera-Correa and colleagues, the authors report an intriguing association between anti-phosphotidyl serine antibodies, FcRL5^+^ T-bet^+^B cells, and hemoglobin levels among patients presenting with symptomatic malaria. The authors further describe in vitro experiments detailing that FCRL5^+^ T-bet^+^ B cells expand after coculture of naïve PBMC with Pf lysate, and that these expanded cells appear to produce anti-PS antibodies, which strengthens the observations reported and suggest that malarial anemia may be mediated in part by auto-immune antibody production by this B cell subset. The manuscript is clearly written and easy to follow. My major concern with the data presented in this manuscript is that the main results (Figures 1-5) are mainly correlations, and that the correlation results were reported with linear regression. With the small patient numbers (n=19) several of the "significant" correlations may have been driven by 1 or 2 "outliers" with lower hemoglobin, for example. Given the small numbers, nonparametric assessments would have been more appropriate (i.e. Spearman's correlation coefficients as opposed to linear regression).

We thank the reviewer for the insightful comments and detailed review. We believe that the quality of the manuscript has been greatly improved.

As suggested by the reviewer, we have now analyzed the data using nonparametric assessment methods (Spearman’s correlation) in all correlations. All figures have been updated to show the results of the new analysis.

Importantly, in many of the figures, there appears to be 27 data points, but in the Materials and methods, the authors state that there were 19 patients (7 with convalescent samples). Thus, one would surmise that the figures reflect 1 data point from 12 individuals, and two data points from 7 individuals. Assuming that these repeated measurements are correlated, the authors should have performed some analysis to account for repeated measures within individuals (e.g. generalized estimating equations or mixed effects linear regression) since autocorrelation could also be driving these associations.

We have modified the text to clarify that the analysis was performed with 31 unique samples from 24 patients (Results section). We have performed an analysis of the repeated samples by Spearman’s correlation method, finding that there is no correlation in atypical MBCs between repeated measurements. This is now mentioned in Materials and methods section.

One possible way to strengthen the authors' claims would be to obtain and analyze samples from a secondary cohort (e.g., an endemic cohort, if obtaining samples from a traveler's clinic is too time consuming.) The authors defend the choice of using a traveler's cohort in the discussion, since in many instances this might represent a primary infection, although I think it would also be very interesting to see if frequencies of FCRL5^+^ T-bet^+^ B cells correlate with anti-PS antibodies in endemic settings as well.

We agree that this comparison would be very interesting, however, we do not have access to the large number of PBMC that are required for this type of analysis from other cohorts in endemic areas since primarily children are affected, and only small volumes of blood can be drawn.

Materials and methods: 19 patients and 4 controlsResults section – 27 samples? This is misleading. Would state X patients, 27 unique samples. (but isn't it actually 20 patients? Your numbering is from 100-119, so you would also count patient 0?)

This has been corrected in the Results section. It is 24 patients (thank you for noticing this error) and a total of 31 samples.

Concern in Figure 1A: 2 patients with Hb <5 but not in table? What are these 2 measurements? Assuming this reflects Hb measured in all patients at the time that parasitemia was assessed (?Day 0) This data should be included in Table 1.

There were no patients with Hb<5. This was a typing error in this particular graph. No other graph included these.

Table 1 generally is difficult to follow and should be clarified (potentially in legend or with updated column headers)– Days since treatment start – needs clarification that this is the day of PBMC/plasma/Hb sampling.– Also for 7 individuals, the 2nd visit is also detailed in the table. I would argue that this information should be on the same "row" in columns to the right rather than a separate row.– Hb measurement is at the date of sampling – do you also have hemoglobin at date parasitemia measured? (as above)– Why does parasite count in Table 1 oscillate between% and parasites/ul? Not easy to interpret this in the table. Presumably these were all converted to parasites/µl for use in Figure 1A. Would use one standard metric here.

Table 1 has been modified to clarify that it shows the “Days since treatment start to sampling”. Each line represents a collected sample, when more than one sample was collected from the same patient two lines with the same patient ID are shown in the table (data cannot be compressed into one single line, since there are two values for several parameters). Hemoglobin levels at day of presentation are now included. Parasitemias are all now expressed as parasites/µl.

Results section the authors state that hemoglobin does not correlate with parasitemia, but this is an incorrect statement – they are not "significantly" correlated, although, visually, a positive correlation is suggested.

This is now corrected in the new version of the manuscript

Results section: "expression of T-bet in CD27^-^CD21^-^FcRL5^+^ B-cells is directly correlated". Assuming that you assessed co-expression of T-bet in FCRL5^+^ cells? Would be a more direct analysis vs. showing the correlation.

As suggested by the reviewer, we have used the double expression of T-bet and FcRL5 to define atypical MBCs (Figure 3A). We have used these markers to define atypical MBCs throughout the manuscript.

We have also analyzed the differential expression of T-bet in classical MBCs and FcRL5^+^ cells, finding that even if there is some expression of T-bet in classical MBCs, the expression of this marker is significantly higher in FcRL5^+^ cells. This is now included in Results and Figure 4—figure supplement 2.

DiscussionParagraph one: you didn't actually test the hypothesis that malaria-associated anemia is mediated by an autoimmune anti-PS response – you tested the correlation per se, which would be consistent with a causal relationship. Would be a little more judicious in your discussion/conclusions.

Corrected in the new version of the manuscript.

[Editors' note: further revisions were requested prior to acceptance, as described below.]

The manuscript has been improved but a few remaining issues, mainly editorial in nature, need to be addressed before acceptance. The issues to consider for revision are outlined below:1) The introduction to each section fails to capture and integrate narratively the flow of the results. Please make a few editorial edits to better highlight this flow and integration. For example, the first sentence of the second section of the Results reads "Our first aim was..". However, the first section of the Results section seems to be the first aim, which is to establish that autoantibodies correlate with anemia and erythrocyte lysis in malaria patients. Similarly, the first sentence of the third section reads, "Our main goal".. Please reword to simply state, "We next sought to determine whether B cell subsets described above correlate with hemoglobin levels.

These have been corrected in the new version of the manuscript.

2) Gating plots and analysis of Figure 3 and Figure 4. AS written, it seems as though the analysis reported in section 2 (Figure 3) and section 3 (Figure 4) use different gating strategies to define populations of atypical memory B cells. Hopefully this is not the case. For example, in section 2, it states that AtMBC were defined as CD19^+^, FCRL5^+^, T-bet^+^. However, in Section 3, it states that AtMBC were defined as CD19^+^CD21^-^CD27^-^FCRL5^+^Tbet^+^. Please 1) integrate the two "definitions" into one and present these in section 2 of the results, with one integrated gating plot included (not separated into one in Figure 3 and one in Figure 4 —figure supplement.Relatedly- gating figure (Figure 4—figure supplement). Not a very convincing plot of atypical memory B cells as cd27^-^cd21^-^ cells (2.9%?). Do you have a more representative gating figure that can be included instead? (especially if your median in tourists is close to 9%., and given the beautiful plot shown in Figure 3?).

Thanks for noticing this. AtMBC were defined as CD19^+^, FcRL5^+^, T-bet^+^ throughout the manuscript. In section 2, we have removed these (CD27^-^ CD21^-^) from the gating definition for atypical MBCs in subsection “Atypical memory FCRL5^+^ T-bet^+^ 180 B-cells correlate with hemoglobin levels in *P. falciparum*-infected returned travelers” and in Figure 4—figure supplement 1).

3) Results subsection “Anti-parasite antibodies do not correlate with FcRL5^+^ T-bet^+^ atypical memory B-cells in P. falciparum-infected patients” header. Response to only one parasite antigen was measured, so this section header should be changed to something similar to the Figure 6 figure legend.One suggestion:"Atypical memory B-cell frequencies do not significantly correlate with anti-PfEBA antibodies in P. falciparum infected patients".

This has been modified following reviewer suggestion in the new version of the manuscript.

4) In Discussion, paragraph seven, there are a duplication in text the correct text needs to be included.

The duplication has been deleted.